# Microglial motility is modulated by neuronal activity and correlates with dendritic spine plasticity in the hippocampus of awake mice

Felix Christopher Nebeling[1]*, Stefanie Poll[1], Lena Christine Justus[1], Julia Steffen[1], Kevin Keppler[2], Manuel Mittag[1], Martin Fuhrmann[1]*

[1]Neuroimmunology and Imaging Group, German Center for Neurodegenerative Diseases, Bonn, Germany; [2]Light Microscopy Facility, German Center for Neurodegenerative Diseases, Bonn, Germany

**\*For correspondence:**
felix.nebeling@dzne.de (FChristopherN);
martin.fuhrmann@dzne.de (MF)

**Competing interest:** The authors declare that no competing interests exist.

**Abstract** Microglia, the resident immune cells of the brain, play a complex role in health and disease. They actively survey the brain parenchyma by physically interacting with other cells and structurally shaping the brain. Yet, the mechanisms underlying microglial motility and significance for synapse stability, especially in the hippocampus during adulthood, remain widely unresolved. Here, we investigated the effect of neuronal activity on microglial motility and the implications for the formation and survival of dendritic spines on hippocampal CA1 neurons in vivo. We used repetitive two-photon in vivo imaging in the hippocampus of awake and anesthetized mice to simultaneously study the motility of microglia and their interaction with dendritic spines. We found that CA3 to CA1 input is sufficient to modulate microglial process motility. Simultaneously, more dendritic spines emerged in mice after awake compared to anesthetized imaging. Interestingly, the rate of microglial contacts with individual dendritic spines and dendrites was associated with the stability, removal, and emergence of dendritic spines. These results suggest that microglia might sense neuronal activity via neurotransmitter release and actively participate in synaptic rewiring of the hippocampal neural network during adulthood. Further, this study has profound relevance for hippocampal learning and memory processes.

## Editor's evaluation

This work provides insights into the use of anesthetics in measuring cellular dynamics, regional differences in plasticity, and neuronal activity regulation of microglia dynamics. It is an important contribution to understanding hippocampal microglia plasticity in adulthood.

## Introduction

Microglia are cells of the innate immune system in the central nervous system (CNS). Under physiological conditions, microglia are sessile, highly ramified, and plastic cells that constantly scan their environment with fine processes, an activity referred to as microglial motility (*Davalos et al., 2005*; *Nimmerjahn et al., 2005*). This remarkable degree of cellular plasticity enables microglia to detect and quickly react to even the subtlest changes of tissue homeostasis within their microenvironment – a process shown to be ATP-dependent (*Davalos et al., 2005*). While microglia have been linked to a number of neurodegenerative diseases such as Alzheimer's disease (*Heneka et al., 2014*; *Ransohoff, 2016*), little is known about their physiological function under healthy conditions (*Tay et al.,*

*2017*; *Tremblay et al., 2011*). During the discovery of microglial motility, the hypothesis was tested as to whether process motility related to neuronal activity, yet interfering with neuronal activity in vivo using tetrodotoxin (TTX) and bicuculline led to inconsistent findings (*Nimmerjahn et al., 2005*). These results may have been confounded by the use of anesthetics like isoflurane, although there is no current consensus on the impact of anesthesia on microglial motility in vivo.

Under fentanyl/midazolam/dexmedetomidine (*Stowell et al., 2019*) or isoflurane (*Liu et al., 2019*) anesthesia in vivo, cortical microglia increased their process area and surveillance territory. However, contradictory effects have been described while using isolfurane anesthesia, showing decreased microglia motility in response to isoflurane in acute slice preparations (*Madry et al., 2018*). Further, a recent study also described microglia motility being positively correlated to neuronal activity in vivo in the cortex (*Hristovska et al., 2022*). These diverging findings might in part be due to different means of preparation, specimen age, and/or brain region specificity, as microglia display a high degree of region-dependent diversity (*Grabert et al., 2016*). Evidence from Ca$^{2+}$ imaging in zebrafish suggests a reciprocal relationship between microglial motility and neuronal activity, with decreased Ca$^{2}$-transient frequencies upon microglial contact with neurons (*Li et al., 2012*). Further evidence for a dependency of microglial process motility on neuronal activity has been derived from studies investigating the relationship of dendritic spine formation and elimination using two-photon in vivo imaging. Lowering neuronal activity by inducing hypothermia and applying TTX reduced the frequency of microglial contacts with dendritic spines in the visual and somatosensory cortex, although microglial motility was not directly measured in this particular study (*Wake et al., 2009*). Moreover, in the visual cortex microglial motility was reduced in mice kept in darkness, whereas it increased in mice re-exposed to light after a dark period (*Tremblay et al., 2010*). The same study found that microglial contact rate increased before dendritic spines were eliminated during a critical period of developmental plasticity, suggesting a role for microglia in shaping post-synapses. During early development in the mouse cortex, microglial contacts at dendrites also induced the formation of new dendritic filopodial spines in awake, but not anesthetized mice (*Miyamoto et al., 2016*). Additionally, removal of microglia from the brain of adult mice reduced motor learning-induced/BDNF-dependent formation of new dendritic spines in the motor cortex (*Parkhurst et al., 2013*).

An important brain region involved in learning and memory processing is the hippocampus. Whether hippocampal microglia have similar functions in dendritic spine formation and elimination as in the cortex remains unresolved. Yet, recent data from in vitro studies showed a relationship of long-term potentiation and spine contact rates with microglia in the hippocampus (*Pfeiffer et al., 2016*). Further, pre-synapses – but not post-synapses – were eliminated by trogocytosis (non-apoptotic uptake of membrane components by immune cells), while filopodial spine formation was induced by microglia during development in hippocampal slice cultures (*Weinhard et al., 2018*). Additional in vitro experiments indicated that microglial responses to neuronal stimulation in the hippocampus were only detectable during early post-natal stages and absent during adulthood (*Logiacco et al., 2021*). Together, these results suggest a role for microglia in promoting synapse stability and formation.

The interplay between microglia and synapses is of particular interest, as microglia are by far the most plastic cells in the CNS. By engaging with synapses, microglia have the potential to enhance and weaken synaptic transmission, as well as affect structural plasticity of spines in healthy and disease conditions. Whereas microglia have been shown to be critical for synaptogenesis and synaptic integrity during developmental stages (*Miyamoto et al., 2016*; *Schafer et al., 2012*; *Stevens et al., 2007*; *Tremblay et al., 2010*), little is known about their contribution to structural plasticity in the hippocampus during adulthood. Thus, it is critical to understand the nature and functional properties of microglia-synapse interactions, especially during adulthood, as the role of microglia in CNS homeostasis might differ between CNS development and adulthood (*Mosser et al., 2017*). Microglial fine process motility has not been examined in vivo in the hippocampus up until now, and recent findings implicate a more heterogeneous role of microglia throughout different brain regions than previously assumed (*Grabert et al., 2016*). It is therefore important to elucidate whether microglial motility in the hippocampus is different from the cortex. For that reason, we set out to investigate microglial motility under healthy conditions and in the absence of anesthesia in the hippocampus of awake, adult mice while examining to what extent baseline surveillance of microglia is dependent on neuronal activity. Additionally, we address the question of whether microglia play a role in dendritic spine formation and elimination during adulthood, supporting lifelong structural plasticity of spines in the hippocampus.

## Results

### Microglial motility correlates with neuronal activity in hippocampal CA1

To address the relationship of neuronal activity and microglial motility in the hippocampus in vivo, we used mice expressing green fluorescent protein (GFP) in microglia (knock in of *gfp* into the *Cx3cr1* [chemokine C-X3-C motif receptor 1] locus [*Jung et al., 2000*]) and yellow fluorescent protein (YFP) in neurons (Thy1-YFP-H mice; *Feng et al., 2000*). Mice were heterozygous for both alleles (*Thy1-YFP$^{tg/wt}$::Cx3cr1-GFP$^{ki/wt}$*) and between 7 and 10 months of age. Of note, *Cx3cr1-GFP$^{ki/wt}$* microglia are haploinsufficient for the fractalkine receptor (CX3CR1). However, in an elaborated transcriptome analysis of mice with complete loss or haploinsufficiency of CX3CR1 compared to the wildtype (WT) condition, major differences were found at a younger age (2 months), whereas WT and heterozygous mice cluster closer together in older mice 1–2 years of age (*Gyoneva et al., 2019*). To access the hippocampus, we unilaterally removed parts of the somatosensory cortex and implanted a metal tube, sealed with a glass cover-slip at the bottom (*Gu et al., 2014*). After at least 4 weeks of recovery, two-photon in vivo imaging began (*Figure 1a and b*). Microglial fine process motility was consecutively recorded (with 3-week intervals between imaging experiments/treatment conditions) in the dorsal hippocampus of the same mice under anesthesia, then in awake conditions running on a circular treadmill, and finally in awake conditions with topical TTX application (*Figure 1a*). Microglia motility was measured by acquiring z-stacks spanning 100 µm in the stratum radiatum (SR) of the dorsal hippocampus (*Figure 1b*). Microglial process turnover rate was calculated as previously described (*Fuhrmann et al., 2010*). The lost and gained proportion of microglia processes between imaging time points was calculated in a two-channel color overlay image (*Figure 1c*; Methods). Under isoflurane anesthesia (1.0% isoflurane in oxygen), microglia showed reduced fine process motility compared to the awake condition (60.2% ± 2.7% anesthesia vs. 69.9% ± 2.8% awake; *Figure 1d*; *Figure 1—videos 1 and 2*). Isoflurane acts primarily by prolonging GABAergic transmission, thus inhibiting neuronal activity (*Goltstein et al., 2015*; *Noda and Takahashi, 2015*; *Vahle-Hinz et al., 2007*). We confirmed that 1% isoflurane anesthesia significantly reduced Ca$^{2+}$ event rate as a marker of neuronal activity by more than 50% in hippocampal CA1 neurons (*Figure 1—figure supplement 1*, *Figure 1—videos 3 and 4*). These results led to the hypothesis that microglial motility and neuronal activity are connected. To address this question, we topically applied TTX (50 µM), a potent blocker of voltage-gated sodium channels, to inhibit action potential firing in the hippocampus. This pharmacological manipulation significantly decreased microglial motility in awake mice, even below the levels of these same mice under anesthesia (*Figure 1d*; 69.9% ± 2.8% awake vs. 52,8±1.2% awake plus TTX). These results suggest a positive relationship between microglial motility and neuronal activity in the dorsal CA1 region of the hippocampus in vivo.

### Modulating CA3 to CA1 input is sufficient to change microglial motility in CA1

In order to further investigate the relationship between microglial motility and neuronal activity, we carried out neuronal silencing experiments with chemogenetic tools. First, we aimed to exclude the effect of TTX application directly on microglial sodium channels, independent of neuronal activity modulation. Under physiological conditions, microglia express a limited number of voltage-gated sodium channels (*Persson et al., 2014*), however a direct effect of TTX on microglia could not be excluded entirely in the previous experiment.

Thus, we sought to simultaneously silence CA1 pyramidal neurons and their inputs from CA3 neurons via the Schaffer collaterals (*Figure 2—figure supplement 1a*) by injecting two adeno-associated viruses (AAVs) to target these neuronal populations – one expressing Cre-recombinase under the neuron-specific Ca$^{2+}$/calmodulin-dependent protein kinase II (CamKII) promoter (AAV9-CamKII-Cre), and another expressing a loxP-flanked version of an inhibitory G-protein coupled receptor (hM4D(Gi)) under the neuronal synapsin I promoter tagged with the fluorescent reporter mCherry (AAV2-hsyn-DIO-hM4D(Gi)- mCherry, *Krashes et al., 2011*). The inhibitory G-protein coupled receptor hM4D(Gi) and the excitatory G-protein coupled receptor hM3(Gq) are DREADDs (designer receptor exclusively activated by designer drugs), that upon binding of the drug clozapine N-oxide (CNO), lead to silencing or activation of transfected neurons, respectively (*Armbruster et al., 2007*; *Stachniak et al., 2014*). DREADDs are routinely used as a tool to alter neuronal activity (*Zhu and Roth, 2014*). We confirmed inhibitory DREADD expression by post hoc immunohistological stainings of transfected hippocampi

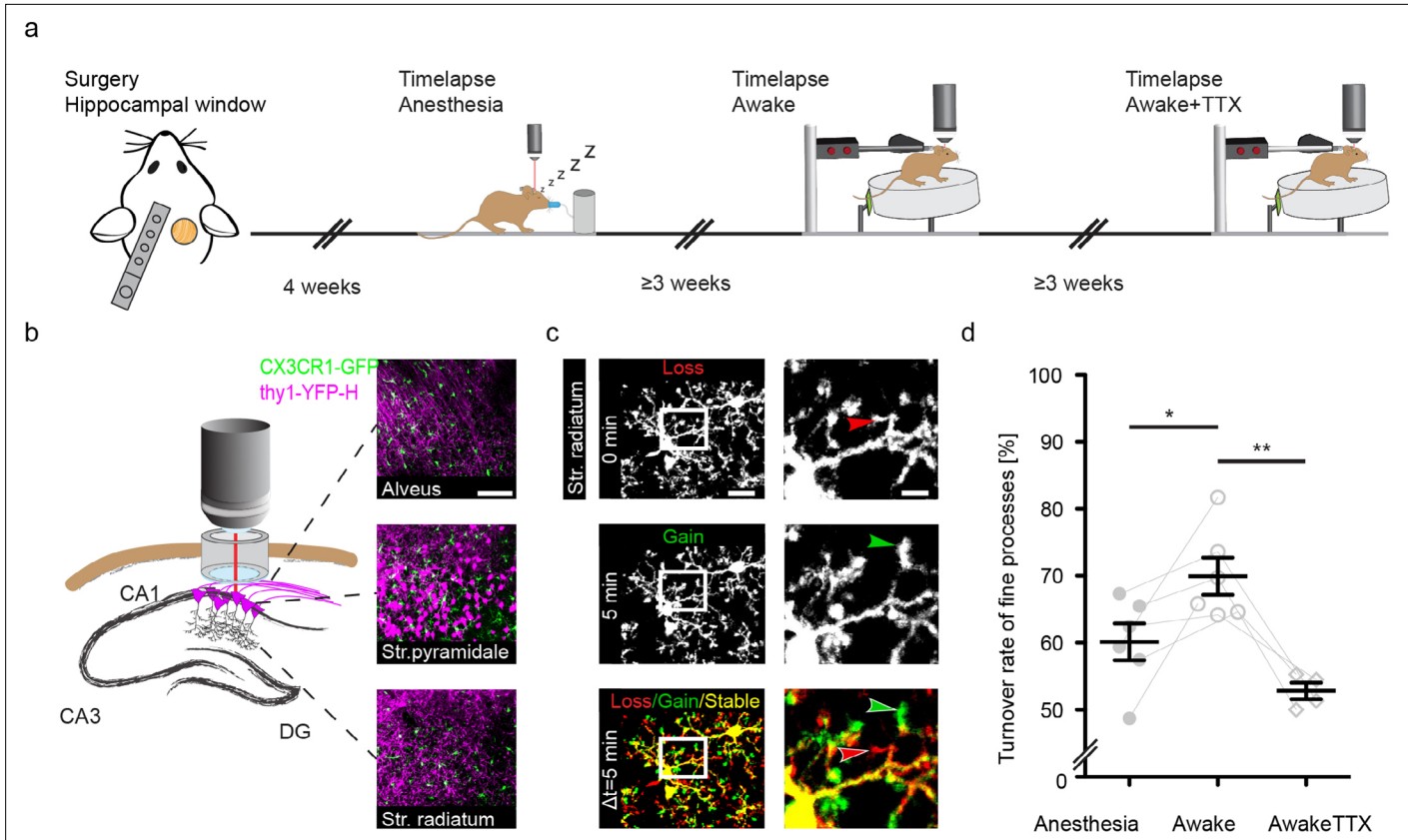

**Figure 1.** Increased microglial motility in the hippocampus of awake mice. Experimental paradigm of repetitive microglia imaging in dorsal CA1 (stratum radiatum [SR]) in the same mice under varying conditions. (**b**) Schematic of the accessible hippocampal layers in CA1 through a chronic hippocampal cranial window. CX3CR1-GFP::thy-1-YFP-H transgenic mice allowed for simultaneous visualization of microglia (green) and neurons (magenta). (**c**) Representative image of a microglia cell and its processes in an awake mouse imaged in CA1, SR. Subsequent time points (0, 5 min) were superimposed (Δt=5 min) to measure gained (green arrow), lost (red arrow), and stable (yellow area) processes. (**d**) Turnover rate of microglia fine processes (i.e. motility) at varying conditions (n=6 mice 'anesthesia', 'awake'; n=4 'awake+tetrodotoxin [TTX]'); one-way ANOVA with Bonferroni's multiple comparison test, $F_{(2,13)} = 10.22$; p=0.0291 (anesthesia vs. awake), p=0.0014 (awake vs. awake+TTX). Error bars: SEM. Scale bars: (**b**) 100 μm, (**c**) 10 μm, 2 μm; *p<0.05, **p<0.01.

The online version of this article includes the following video and figure supplement(s) for figure 1:

**Figure supplement 1.** Isoflurane anesthesia decreases neuronal activity in the hippocampus.

**Figure supplement 2.** Examples of microglia in CA1 under anesthesia, awake, and tetrodotoxin (TTX)-treated conditions.

**Figure 1—video 1.** Microglial motility in an anesthetized mouse.

https://elifesciences.org/articles/83176/figures#fig1video1

**Figure 1—video 2.** Microglial motility in an awake mouse.

https://elifesciences.org/articles/83176/figures#fig1video2

**Figure 1—video 3.** Calcium imaging of CA1 pyramidal neurons under isoflurane anesthesia.

https://elifesciences.org/articles/83176/figures#fig1video3

**Figure 1—video 4.** Calcium imaging of CA1 pyramidal cells in an awake mouse.

https://elifesciences.org/articles/83176/figures#fig1video4

---

(*Figure 2—figure supplement 2b–d*). We analyzed microglial motility in the same awake mice with and without CNO application (*Figure 2—figure supplement 1e, f*). CNO-mediated silencing of CA3 and CA1 neurons significantly reduced microglial motility (*Figure 2—figure supplement 1g*; 71.2%±1.5% vehicle vs. 58.7%±1.8% CNO).

Next, we specifically silenced or activated CA3 input to CA1 and measured microglial motility in CA1 of awake mice (*Figure 2*). As before, we injected two AAVs – one expressing Cre-recombinase under the neuron-specific Ca²⁺/CamKII promoter (AAV9-CamKII-Cre, Addgene), and another

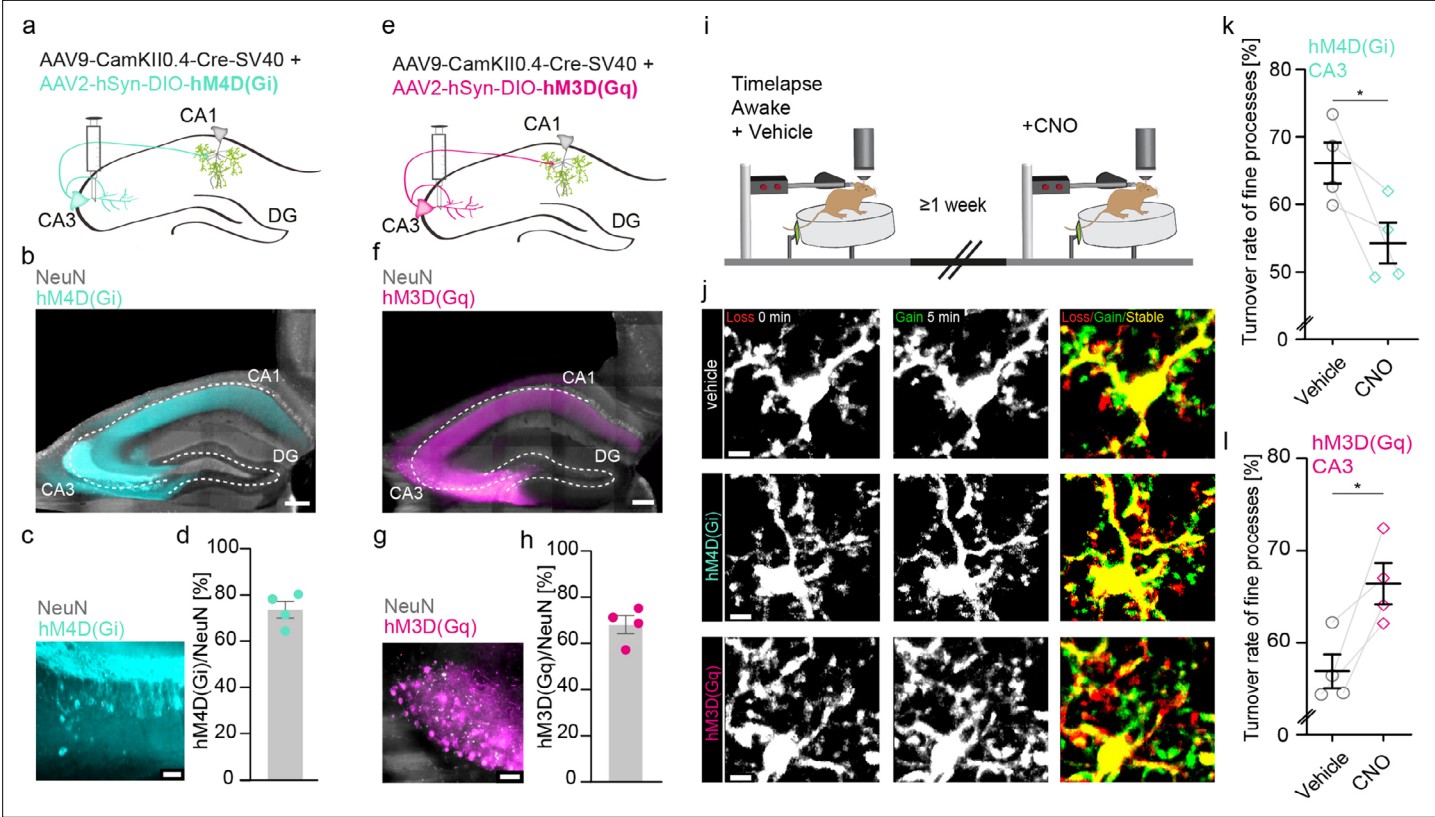

**Figure 2.** Increasing or decreasing CA3 input to CA1 reciprocally changes microglial motility. (**a, e**) Adeno-associated virus (AAV) injections into the CA3 region of the hippocampus with CamKII-driven Cre-recombinase expression in neurons. This allows for the expression of loxP-flanked hM4D(Gi) (**a**) or hM3D(Gq) (**e**) under human synapsin1 promoter in CA3 neurons. (**b, c**) Representative images of hM4D(Gi) expression in the hippocampus (**b**), zoom on CA3 (**c**). (**d**) Quantification of hM4D(Gi)-DREADD (designer receptor exclusively activated by designer drug) transfected cells, on average 74% of cells in CA3 were co-expressing NeuN and mCherry/hM4D(Gi) (n=4 mice, 2410 cells were counted in total). (**f, g**) Representative images of hM3D(Gq) expression in the hippocampus (**f**), zoom on CA3 (**g**). (**h**) Quantification of hM3D(Gq)-DREADD transfected cells, on average 68% of cells in CA3 were co-expressing NeuN and mCherry/hM3D(Gq) (n=4 mice, 1749 cells were counted in total). (**i**) The dorsal CA1 region (stratum radiatum [SR]) of the same mice was imaged awake twice with a 1-week interval. Imaging started 40 min after i.p. injection of either vehicle (dimethyl sulfoxide [DMSO]) or clozapine N-oxide (CNO). (**j**) Exemplary images of microglia fine process turnover after vehicle and CNO injections under both hM4D(Gi) and hM3D(Gq) conditions in CA1, SR. (**k**) Microglia fine process turnover rate of DMSO- and CNO-treated mice that were injected with an inhibitory DREADD, hM4D(Gi) (n=4 mice, unpaired t-test, two-tailed, p=0.0326). (**l**) Microglia fine process turnover rate of DMSO- and CNO-treated mice that were injected with an excitatory DREADD, hM3D(Gq) (n=4 mice, unpaired t-test, two-tailed, p=0.0167). Error bars: SEM. Scale bars: (**j**) 5 µm, (**c,g**) 50 µm, (**b,f**) 500 µm, 2 µm; *p<0.05, ***p<0.001.

The online version of this article includes the following video and figure supplement(s) for figure 2:

**Figure supplement 1.** Microglial motility in awake mice depends on neuronal activity.

**Figure supplement 2.** Elevated Ca$^{2+}$ event frequency in Schaffer collaterals following DREADD (designer receptor exclusively activated by designer drug) activation.

**Figure 2—video 1.** Ca$^{2+}$ imaging of Schaffer collaterals in CA1 of vehicle-treated hM3D(Gq)-expressing CA3 neurons in awake mice.
https://elifesciences.org/articles/83176/figures#fig2video1

**Figure 2—video 2.** Ca$^{2+}$ imaging of Schaffer collaterals in CA1 of clozapine N-oxide (CNO)-treated hM3D(Gq)-expressing CA3 neurons in awake mice.
https://elifesciences.org/articles/83176/figures#fig2video2

loxP-flanked version of either an inhibitory or excitatory G-protein coupled receptor (hM4D(Gi)/ hM3(Gq)) under the neuronal synapsin I promoter tagged with the fluorescent reporter mCherry AAV2-hsyn-DIO-hM4D(Gi)-mCherry (*Krashes et al., 2011*), AAV2-hSyn-DIO-hM3(Gq) into the CA3 region of the hippocampus (*Figure 2a and e*). DREADD expression was quantified by post hoc immunohistological stainings of transfected hippocampi. In respective CA3 regions, 74% of neurons expressed hM4D(Gi) (*Figure 2b–d*) or 68% of neurons expressed hM3D(Gq) (*Figure 2f–h*). To validate the DREADD system, we co-injected AAV9-CamKII-Cre alongside AAV1.Syn.Flex.GCaMP6m

and AAV2-hSyn-DIO-hM3D(Gq) (*Figure 2—figure supplement 2*). This facilitates the visualization of Schaffer collaterals in SR and the measurement of CA3 axonal $Ca^{2+}$ event frequency in awake head-fixed mice (*Figure 2—figure supplement 2a–f*). CNO-mediated activation of hM3D(Gq)-expressing CA3 neurons significantly increased axonal $Ca^{2+}$ event frequency compared to vehicle injections, suggesting effective DREADD expression and agonist delivery (*Figure 2—figure supplement 2g*; *Figure 2—videos 1 and 2*). We then analyzed microglial motility in awake mice with and without CNO-mediated silencing or activation of CA3 input to CA1 (*Figure 2i*). Inhibition of CA3 input to CA1 neurons via the Schaffer collaterals significantly decreased microglial motility in the SR (*Figure 2j and k*; 66.1%±3% vehicle vs. 54.3 ± 3% CNO). Conversely chemogenetic activation of hM3D(Gq) trans-fected CA3 neurons significantly increased microglial motility (*Figure 2j and l*; 56.6% ± 1.8% vehicle vs. 66.6%±2.2% CNO). These data indicate that manipulation of CA3 input to CA1 is sufficient to alter microglial motility in the SR of CA1.

## Spine stability correlates with microglial contact rate

CA3 input to CA1 via Schaffer collaterals is an important pathway of the hippocampus involved in spatial information processing and learning. CA3 axons form synapses with dendritic spines of CA1 neurons in the SR and their stimulation can lead to long-term synaptic plasticity (*Luscher and Malenka, 2012*; *Takeuchi et al., 2014*). Microglia have been shown to increase their process length and prolong their contacts with dendritic spines in hippocampal slices during long-term potentiation in situ (*Pfeiffer et al., 2016*). Since microglial motility was related to CA3 activity, we hypothesized that altered microglial motility would correlate with structural plasticity changes of dendritic spines in vivo. First, we investigated whether any differences existed in structural plasticity of dendritic spines in anesthetized and awake mice (*Figure 3*). To do so, we acquired z-stacks containing radial oblique dendrites in the SR of the hippocampus. Repetitive imaging sessions were carried out, first under anesthesia and then under awake conditions on a treadmill with a 2-day interval between imaging sessions (*Figure 3a and b*). Here, we observed a significant increase in spine density and significantly higher fractions of gained vs. lost spines 2 days after awake imaging (*Figure 3c–f*). These results underscore the structural plasticity of dendritic spines in the hippocampus even during adulthood. Moreover, a substantial fraction of dendritic spine turnover was spatially clustered along the dendrites of CA1 pyramidal neurons (i.e. within 4 µm along the same dendrite; 55% of observed turnover, see *Figure 3—figure supplement 1a-c*). Similar findings have previously been reported for dendrites of excitatory neurons in the hippocampus, somatosensory, motor, visual, and retrosplenial cortices (*Bloss et al., 2018*; *Chen et al., 2012*; *Frank et al., 2018*; *Fu et al., 2012*).

After identifying altered structural plasticity of dendritic spines in awake and anesthetized mice, we next investigated whether microglia were involved in synaptogenesis and synapse elimination. This has been shown in other organisms and brain regions in vivo, and the hippocampus in vitro (*Miyamoto et al., 2016*; *Parkhurst et al., 2013*; *Sipe et al., 2016*; *Stevens et al., 2007*; *Tremblay et al., 2010*; *Weinhard et al., 2018*). Here, we analyzed the interaction of microglia with individual dendritic spines and shafts in the SR of the hippocampus in vivo (*Figure 4—video 1*). Theoretically, after being contacted by microglia, a dendritic spine might remain stable, be lost, or conversely, microglial dendrite contact may be followed by new spine formation (*Figure 4a*). To investigate these possibilities, we carried out one 45 min time-lapse imaging (5 min frame rate), followed by another imaging session 2 days later to correlate microglial contact rate with subsequent loss or gain of dendritic spines (*Figure 4b*). Indeed, we observed both spine loss (*Figure 4c–e*) and spine formation (*Figure 4f–h*) linked to prior microglia-dendrite contact. Relative to anesthetized conditions, microglia had significantly higher contact rate with dendritic spines under awake conditions (*Figure 4i*; 5.31±0.16 $hr^{-1}$ in anesthesia vs. 7.33±0.21 $hr^{-1}$ awake). However, there was no significant difference in contact with the dendritic shaft by microglia between awake and anesthetized conditions (*Figure 4j*; 5.55±0.16 $hr^{-1}$ in anesthesia vs. 6.22±0.36 $hr^{-1}$ awake).

Next, we investigated whether the fate of spines (gained, lost, stable) was influenced by the microglial contact rate in the preceding imaging session. Interestingly, the contact frequency of both gained (*Figure 4k*; stable 5.14±0.16 $hr^{-1}$ vs. gained 9.93±0.15 $hr^{-1}$) and lost (*Figure 4k*; stable spines: 5.31±0.16 $hr^{-1}$ vs. lost spines 8.37±1.12 $hr^{-1}$) spines was significantly increased compared to stable spines, indicating that microglial contact frequency is positively associated with structural plasticity of dendritic spines. Moreover, microglial contacts were more frequent at sites on the dendritic shaft

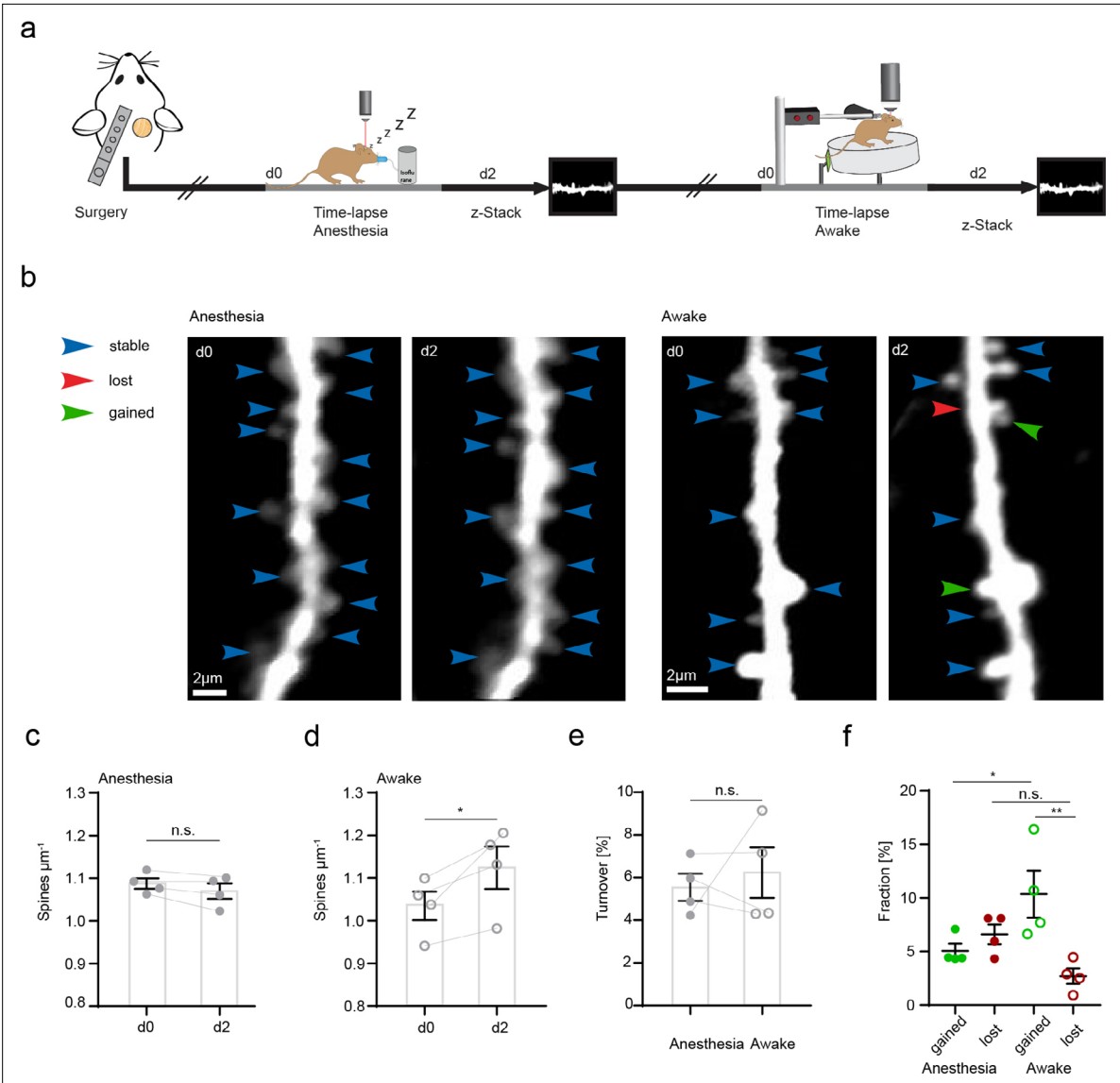

**Figure 3.** Increased structural plasticity of dendritic spines in the hippocampus of awake mice. (**a**) Mice were imaged twice with a 2-day interval, first in anesthesia and subsequently awake on a spinning disk. (**b**) We acquired z-stacks containing radial oblique dendrites in stratum radiatum of dorsal CA1 in the hippocampus. Exemplary images of two dendrites acquired with a 2-day interval in anesthesia (left panel) or awake (right panel). Blue arrows indicate stable, red arrows lost, and green arrows gained spines. (**c**) No changes of spine density after imaging in anesthesia (1.09 ± 0.01 µm⁻¹ at d0 vs. 1.07 ± 0.02 µm⁻¹ at d2; paired t-test, p=0.1235). (**d**) Elevated spine density after awake imaging (1.04 ± 0.03 µm⁻¹ to 1.12±0.05 µm⁻¹; paired t-test, p=0.0462). (**e**) Similar turnover of dendritic spines was detectable between anesthetized and awake mice (5.55 ± 0.64% anesthesia vs. 6.23 ± 1.17% awake; paired t-test, p=0.6692). (**f**) Turnover of dendritic spines differentiated into fractions of lost and gained spines under both conditions. In awake mice, the fraction of gained spines was elevated (10.35 ± 2.19% in awake mice and 5.06 ± 0.68% in anesthetized mice; p=0.039). The fractions of gained and lost spines were comparable under anesthetized conditions (gained: 5.06 ± 0.68% vs. lost: 6.61 ± 0.92%; p=0.4095). The fraction of lost spines was similar in awake compared to anesthetized mice (**f**; 2.7 ± 0.73% awake vs. 6.61 ± 0.92% anesthesia; p=0.1496), under awake conditions we found more gained compared to lost spines (gained: 10.35 ± 2.19% and lost 2.7 ± 0.73%; p=0.0037), ordinary one-way ANOVA followed by Tukey's multiple comparisons test, F(3,12) = 6.23. Data from four mice; n.s. not significant, *p*P*<0.05, **p<0.01. Error bars: SEM.

The online version of this article includes the following figure supplement(s) for figure 3:

**Figure supplement 1.** Clustered structural plasticity of spines associated with microglia contact frequency.

**Figure supplement 2.** Increased structural plasticity of dendritic spines in the hippocampus of awake mice, statistics over dendrites.

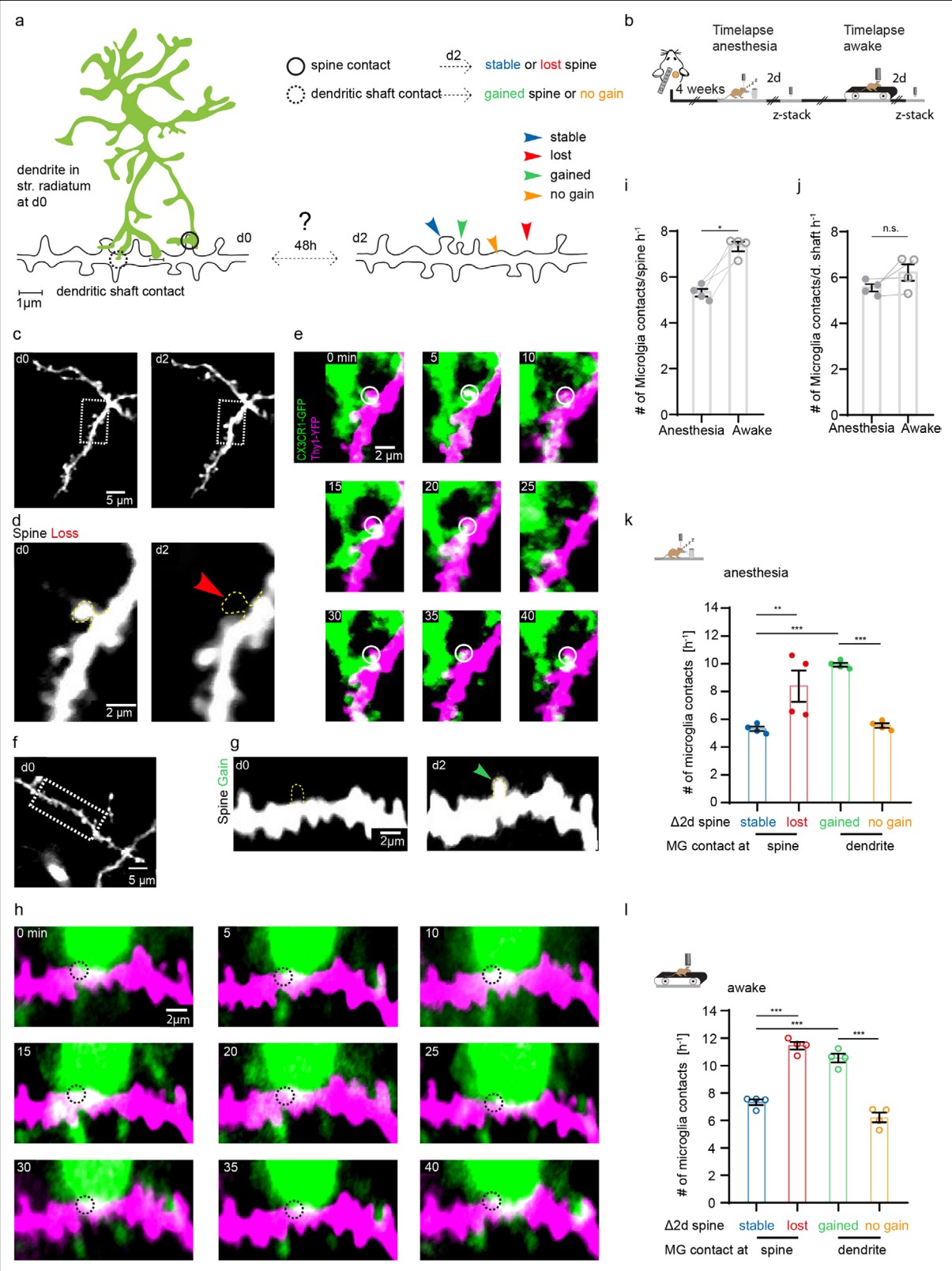

**Figure 4.** Microglia contact frequency is associated with spine stability in the hippocampus. (**a**) Schematic illustrating the analysis between microglia contacts and spine stability in stratum radiatum of dorsal CA1: direct spine contacts (circle) can lead to a stable (blue arrow) or lost spine (red arrow) as analyzed 2 days later, whereas contacts of the dendritic shaft (dashed circle) can lead to a gained spine (green arrow) or no change/no spine gain (orange). (**b**) The same mice were imaged four times after implantation of the hippocampal window. First in anesthesia and 48 hr later two visualize

*Figure 4 continued on next page*

Figure 4 continued

potential changes of dendritc spines and then again under awake conditions, followed by relocalization 48 hr later. (**c**) Overview image of a dendrite in stratum radiatum (SR) at two consecutive time points 48 hr apart. (**d**) Zoom image from (**c**) illustrating a dendritic segment with distinctive dendritic spines. Note the lost spine after 2 days (red arrow). (**e**) Time-lapse of microglia processes (green) interacting with the dendrite and spines (magenta) in the hippocampus of an awake CX3CR1-GFP::thy-1-YFP-H mouse; the circle mark individual microglia contacts of spines. (**f–h**) Analog illustration as depicted in c–e for a gained spine example (green arrow in g). (**i, j**) Direct microglia contacts (**h**) and contacts along the dendritic shaft (**i**) in anesthetized vs. awake mice paired t-test (**h, i**); p=0.0105 (**h**), p=0.1452 (**i**). (**k, l**) Contact rate in anesthesia (**j**) and awake (**k**) at dendritic spines that are either stable (blue) or lost (red) and contacts at the dendritic shaft where either a spine was gained (green) after 48 hr or at random locations along the dendrite where no spine gain was detectable (orange). One-way ANOVA with Sidak's multiple comparison test (**j, k**); $F_{(3,12)}=80.8$ (**j**) and $F_{(3,12)}=73.91$ (**k**); spine contacts averaged over n=4 mice (**h–k**); *p<0.05, **p<0.01, ***p<0.001. Error bars: SEM.

The online version of this article includes the following video and figure supplement(s) for figure 4:

**Figure supplement 1.** Microglia contact frequency associated with spine stability in the hippocampus, statistics over dendrites.

**Figure 4—video 1.** Microglia interacting with dendrites and spines in the hippocampus of an awake mouse.

https://elifesciences.org/articles/83176/figures#fig4video1

where spines were gained, compared to dendritic segments where no new spines were detectable (*Figure 4k*; dendritic shafts without spine gain 5.55±0.16 hr$^{-1}$ vs. dendritic shafts with spine gain 9.93±0.15 hr$^{-1}$). These effects were conserved in awake mice, yet the overall frequency of contacts was increased in line with our previous findings (*Figure 4l*; stable spines 7.33±0.21 hr$^{-1}$, lost spines 11.45±0.27 hr$^{-1}$, gained spines 10.55±0.31 hr$^{-1}$, dendritic shaft 6.22±0.36 hr$^{-1}$). Finally, microglial contacts of dendritic spines were significantly higher where spine turnover occurred in clusters (*Figure 3—figure supplement 1d*), suggesting microglia might facilitate spatial reorganization of dendritic spines in the hippocampus.

In summary, these data show that microglial contact frequency of spines was increased in mice under awake relative to anesthetized conditions, corresponding to increased microglial motility in response to CA3 input/neuronal activity. Furthermore, the turnover (gain and loss) of dendritic spines correlated with higher microglial contact frequency on hippocampal CA1 neurons of adult mice.

## Discussion

In this work we set out to investigate different functional properties of microglia in the hippocampus of mice in vivo, by means of two-photon microscopy. We detected elevated motility of microglia fine processes in awake mice compared to the same mice under anesthetized conditions. We further demonstrated a modulation of microglial motility via CA3 to CA1 input using both pharmacological and chemogenetic approaches. Further, we elucidated the role of microglia in maintaining or destabilizing synapses through an association between microglial contacts of dendritic spines and their stability. Here, lost spines and dendritic stretches with a newly appearing spine were contacted more frequently by microglia in a preceding imaging session than stable spines. Reducing neuronal activity by anesthesia reduced microglia-spine interaction, but did not affect the aforementioned relationship between contact and turnover. Furthermore, clustered events of spine gain or loss were associated with higher microglial contact frequency. These results form the basis for future work unraveling whether microglia-mediated synaptic structural plasticity is involved in hippocampus-dependent behavior, such as spatial navigation, learning, and memory.

### Microglial process motility

Recent findings indicate that there is considerable effect on microglia by the volatile narcotic isoflurane via THIK-1 channels, which reduces microglial scanning activity in vitro in the hippocampus (*Madry et al., 2018*). Here, we investigate microglial motility in the hippocampus in vivo, where we observed reduced microglial fine process motility in anesthetized mice compared to awake conditions. Our data suggest that this decreased microglial motility may be due to reduced neuronal activity. Applying TTX to the hippocampus reduced microglial motility in awake mice. However, in previous studies no such effect of TTX on microglial process motility was detectable in hippocampal slices or the cortex in vivo (*Hines et al., 2009*; *Nimmerjahn et al., 2005*; *Wake et al., 2009*). In these previous experiments, neuronal activity might have already been reduced by the use of anesthetics (*Nimmerjahn et al., 2005*; *Wake et al., 2009*) or been altered in the preparation of slice cultures (*Hines et al., 2009*),

where neuronal activity and connectivity is no longer intact. Another difference between past in vivo studies and the current one is the analyzed brain region. Our study has been carried out in the SR of the hippocampus, whereas former studies investigated the somatosensory cortex. Region-specific microglia mRNA expression profiles could contribute to the motility difference between cortex and hippocampus (*Grabert et al., 2016*). Furthermore, TTX is a potent blocker of voltage-gated sodium channels and may directly affect microglia, as they express TTX-sensitive sodium channels. TTX application in vitro reduces microglial phagocytosis and lamellipodia formation (*Black et al., 2009*; *Persson et al., 2014*). To avoid a potential direct modulatory effect of TTX on microglial motility, we chose a chemogenetic approach to specifically silence or excite hippocampal neurons in vivo. In our study, silencing CA3 neurons (and their input to CA1 via Schaffer collaterals) reduced microglial motility in the SR of the CA1. Conversely, increasing the firing rate of CA3 neurons increased microglial motility in the SR. In contrast to our observations in the hippocampus, isoflurane anesthesia increased microglial arborization and motility in the cortex in vivo (*Liu et al., 2019*), while other studies identified increased arborization alone (area coverage by microglia) (*Sun et al., 2019*; *Umpierre et al., 2020*). Stowell et al. showed effects on microglial arborization under a mixture of fentanyl, midazolam, and dexmedetomidine arguing for a noradrenergic regulation of microglial surveillance (*Stowell et al., 2019*). Other non-isoflurane anesthetics had the opposite effect on microglial motility; both ketamine/xylazine or pentobarbital administration decreased microglial motility (*Hristovska et al., 2020*). Thus, consensus on the impact of anesthetics on microglia behaviors has not been achieved, however our data clearly indicate that neuronal activity and neurotransmitter release regulate microglial motility. Different neurotransmitters might have different effects. Whether our and previous findings indicate brain region-dependent diversity of microglia (*Grabert et al., 2016*) remains to be analyzed.

By activating or inactivating neurons in the CA3 region using DREADDs, glutamatergic output is altered. We found that a CNO-mediated increase (hM3Gq) or decrease (hM4Di) in CA3 neuronal activity corresponded to increased (hM3Gq) or decreased (hM4Di) microglial motility in the hippocampus, respectively, suggesting that glutamate may regulate microglial motility in the region. This notion is supported by work showing increased microglial motility in the cortex upon glutamatergic stimulation in slices and in vivo (*Eyo et al., 2014*). The response of microglia to increased extracellular glutamate might be due to a co-release of ATP and subsequent activation of microglial P2YR12 receptors (*Eyo et al., 2014*). Alternatively, there might be an indirect effect of synaptically released glutamate on microglia via astrocytes, that detect glutamate and subsequently release GABA, which in turn acts on microglia via GABA$_B$ receptors (*Logiacco et al., 2021*). The exact molecular mechanism underlying the microglial response to glutamate remains to be determined.

Based on our results, we conclude that microglial motility is modulated by CA3 input to CA1 in adult mice in vivo, potentially via glutamate release from Schaffer collaterals in the SR. It should be noted that baseline motility was not abolished, but attenuated following pharmacological and chemogenetic manipulation, suggesting additional regulatory mechanisms for microglial motility in this context. These findings argue for a relationship of glutamate release and microglial motility in the hippocampus.

## Microglial contact rates of dendritic spines are increased in awake mice

Until now, few studies have investigated changes in hippocampal synaptic connectivity in vivo by analyzing structural plasticity of dendritic spines. The first study to do so investigated structural plasticity of spines in the hippocampus over hours, detecting only minor changes (*Mizrahi et al., 2004*). In comparison, by investigating structural plasticity over weeks, we and others demonstrated synaptic rewiring in adult mice (*Attardo et al., 2015*; *Gu et al., 2014*) given that the majority of dendritic spines carry post-synaptic densities and form synapses (*Cane et al., 2014*; *Runge et al., 2020*; *Trachtenberg et al., 2002*). In addition to excitatory neurons, a subset of inhibitory neurons also exhibit structural plasticity of dendritic spines (*Schmid et al., 2016*), supporting the view that synapses in the hippocampus are constantly emerging or eliminated. This capacity might facilitate hippocampal adaptation to altered external stimuli and/or different environments. The structural plasticity of dendritic spines in the hippocampus might still be underestimated, as recent studies using super-resolution approaches revealed high turnover rates of dendritic spines in this region – although analysis lasted only days to weeks (*Pfeiffer et al., 2018*). It is important to note that all previous studies have been carried out under anesthesia. Here, we demonstrate structural plasticity of dendritic spines on apical dendrites of

CA1 pyramidal neurons in awake mice, offering the potential for future studies to combine this kind of imaging with defined sensory stimuli or behavioral tasks. Indeed, awake mice, meaning imaged under awake conditions while being on a treadmill for 1 hr, exhibited a higher spine density after 2 days compared with anesthetized mice. We suspect the increase observed in our study to be transient, as permanent increases in spine formation were not observed in previous studies (*Attardo et al., 2015*; *Gu et al., 2014*; *Pfeiffer et al., 2018*). Although mice were habituated, it is possible that the 1 hr awake imaging is similar to an enriched environment, which induced a dendritic spine increase in the hippocampus (*Rampon et al., 2000*) and the cortex (*Jung and Herms, 2014*). Additionally, isoflurane, a volatile anesthetic, can influence dendritic spine characteristics and synaptic transmission (*Platholi et al., 2014*; *Yang et al., 2021*). In vitro, isoflurane reduced spine density, likely by reducing F-actin in spines (*Kaech et al., 1999*; *Platholi et al., 2014*), while in vivo isoflurane potentiated glutamatergic synaptic transmission in mice, but did not impact spine density or turnover (*Antila et al., 2017*). However, Yang et al. found that the fraction of gained spines and thus overall spine turnover was slightly increased under repetitive isoflurane anesthesia (*Yang et al., 2021*).

In order to integrate our findings on microglial motility with structural plasticity of dendritic spines, we examined the interaction of microglial processes with dendritic spines. While microglia have traditionally been studied in the context of immune and inflammatory processes (*Deczkowska et al., 2018*; *Prinz and Priller, 2014*), growing evidence shows the importance of microglia for normal CNS maturation, function, and homeostasis (*Mosser et al., 2017*; *York et al., 2018*). A compelling idea is that microglia may actively contribute to structural plasticity of pre- and post-synapses. This hypothesis derives from studies showing the necessity of normal microglia function for synaptic integrity and learning-dependent synapse formation (*Bessis et al., 2007*; *Lim et al., 2013*; *Miyamoto et al., 2016*; *Paolicelli et al., 2011*; *Parkhurst et al., 2013*; *Pfeiffer et al., 2016*; *Reshef et al., 2017*; *Rice et al., 2015*; *Rogers et al., 2011*; *Schafer et al., 2012*; *Sipe et al., 2016*; *Stevens et al., 2007*; *Tremblay et al., 2010*; *Weinhard et al., 2018*). Yet, studies directly monitoring microglia synapse interaction are sparse (*Lowery et al., 2017*; *Miyamoto et al., 2016*; *Pfeiffer et al., 2016*; *Tremblay et al., 2010*; *Wake et al., 2009*; *Weinhard et al., 2018*) and have been performed either in the cortex under anesthesia, during early development, or in slice cultures. Microglia-synapse interaction has not yet been assessed in the hippocampus of adult awake mice. Here, in the hippocampus, we observed higher contact rates of microglia with dendritic spines than was described for cortical areas or slice cultures. We found that the average contact rate of microglia with stable spines was increased in awake compared to anesthetized conditions. The lower microglial contact rates of dendritic spines observed in previous studies may be due to the use of anesthesia or the nature of in vitro approaches, which cannot fully replicate the complexity of the in vivo environment.

## Microglial contact rate associated with stability of dendritic spines

Structural plasticity of dendritic spines involves two main events: (1) the loss of existing spines and (2) the emergence of new spines, called spinogenesis. We investigated both spine loss and spinogenesis in relation to previous microglial contacts. Microglia are important players during early CNS development, particularly for removal of supernumerary synapses (*Paolicelli et al., 2011*; *Schafer et al., 2012*; *Stevens et al., 2007*), but their role in synapse loss in the adult brain is not well understood. Our time-lapse imaging revealed an association between microglial contacts and the loss of dendritic spines. However, whether microglia actively phagocytose dendritic spines, as suggested by previous studies (*Paolicelli et al., 2011*; *Sipe et al., 2016*; *Tremblay et al., 2010*), or induce retraction into the dendrite remains unclear. Interestingly, *Weinhard et al., 2018*, showed in an in vitro model of early development that microglia do not phagocytose post-synaptic synapses, but refine pre-synaptic boutons and axons (*Weinhard et al., 2018*). The mechanism of microglia-mediated spine loss in the hippocampus in vivo remains to be investigated.

We also analyzed the formation of new spines along dendritic shaft segments that received microglial interaction during a preceding imaging session. Similar to the results for spine loss, we found an association between localized dendritic spine formation and prior dendritic contact by microglia, consistent with previous studies showing microglial regulation of synaptogenesis (*Miyamoto et al., 2016*; *Parkhurst et al., 2013*; *Reshef et al., 2017*; *Weinhard et al., 2018*). As microglia display high brain region-dependent diversity (*Grabert et al., 2016*), there is an unmet demand for functional studies of microglia in distinct brain regions, especially in the hippocampus. Here, we provide evidence

for a link between microglia-dendrite interactions and new spine formation in the adult hippocampus. Due to the resolution limit of two-photon microscopy (~400 nm in x,y-plane), all microglial contacts of dendritic spines measured in this study are putative contacts. To ultimately resolve microglia spine contacts, super-resolution imaging (*Fuhrmann et al., 2022*; *Pfeiffer et al., 2018*) or electron microscopy (*Tremblay et al., 2010*) would be necessary.

Mechanisms for microglia-mediated spine loss presumably involve phagocytosis or trogocytosis, as discussed above. However, during our 45 min time-lapse imaging we did not detect a single spine loss or gain, suggesting two separate timelines, where microglial contacts and motility take place on the second to minute timescale (*Lowery et al., 2017*; *Tremblay et al., 2010*) and structural dendritic spine plasticity happens within the subsequent hours to days (*Attardo et al., 2015*; *Gu et al., 2014*; *Mizrahi et al., 2004*; *Pfeiffer et al., 2018*). The mechanisms that reconcile microglial contact rate with structural plasticity of dendritic spines remain to be discovered. In hippocampal brain slices derived from early post-natal, but not adult hippocampi, microglia respond to electrical stimulation of Schaffer collaterals via $Ca^{2+}$ elevations in the somata and processes (*Logiacco et al., 2021*). Potentially the microglial response to increased levels of glutamate might be due to activity-induced glutamate uptake and the subsequent release of GABA by astrocytes; this might function as a relay station between synapses and microglia, triggering a microglial response through the activation of $GABA_B$ receptors on microglia (*Logiacco et al., 2021*). Another potential modulatory candidate of the microglia-synapse interaction is norepinephrine. Upon norepinephrine binding to $\beta_2$-AR, microglial contacts with dendritic spines are significantly reduced in the mouse visual cortex (*Stowell et al., 2019*). While there is some preliminary evidence for the mechanism underlying microglia-mediated spine loss, the mechanism by which microglia might induce spinogenesis upon contacts with the dendritic shaft remains unresolved. In our study, we found that glutamatergic input is sufficient to modulate microglial motility in SR. Here, input from CA3 neurons via the Schaffer collaterals might direct microglial processes toward CA1 dendrites and facilitate structural plasticity. These changes may require subtle, local alterations of actin polymerization, which is the basis for the emergence of new dendritic spines and memory formation (*Basu and Lamprecht, 2018*). A potential link between microglial contacts and spinogenesis is elevated intracellular dendritic $Ca^{2+}$ following microglial contacts, at least during early development (*Miyamoto et al., 2016*), which may drive accumulation of actin (*Oertner and Matus, 2005*). In developing organotypic slice cultures, microglial contacts induced filopodial outgrowth from the spine-head (*Weinhard et al., 2018*). In our study, we demonstrated an association between putative microglial contacts with the dendritic shaft and subsequent spine outgrowth within the adult hippocampus in vivo, supporting microglia-mediated structural plasticity of dendritic spines not only during development, but throughout the entire lifespan. Microglial involvement in structural synaptic plasticity may yield promising interventional approaches for microglial manipulation in neurological diseases, where synaptic plasticity is impaired, including Alzheimer's disease (*Hoffmann et al., 2013*; *Schmid et al., 2016*), major depressive disorder (*Duman et al., 2016*), and schizophrenia (*Druart et al., 2021*).

## Clustering of dendritic spines and microglial contact rates

Activation of clustered synapses generates relatively large post-synaptic potentials, not correlated with the total number of single synapses (*Palmer et al., 2014*; *Wilson et al., 2016*), but instead described by a nonlinear model of signal transduction (*Kastellakis et al., 2015*; *Larkum and Nevian, 2008*), thereby facilitating information transfer and increasing signal propagation efficiency. Memory is commonly theorized to be stored in small and overlapping distributed neuronal populations, in which these synaptic clusters on dendritic branches code for associated information about time, space, and context (*Cai et al., 2016*; *Kastellakis et al., 2015*; *Rashid et al., 2016*; *Rogerson et al., 2014*; *Silva et al., 2009*). Previous studies show preferential turnover of spatially clustered dendritic spines in excitatory cells in the hippocampus, somatosensory, motor, visual, and retrosplenial cortex (*Bloss et al., 2018*; *Chen et al., 2012*; *Frank et al., 2018*; *Fu et al., 2012*). Accordingly, we observed that approximately 55% of dendritic spine turnover was spatially clustered. Microglia-contacted spines emerged or disappeared in clusters more frequently than single gained or lost spines, suggesting microglial involvement in clustered turnover of hippocampal dendritic spines.

## Limitations

Finally, we would like to address the limitations of of this study. First, CX3CR1-GFP mice were haploinsufficient for the *Cx3cr1* gene, which means that our findings cannot be transferred to WT animals. Another potential caveat is the hippocampal window implantation procedure, which requires removal of cortical tissue. To our knowledge, no method exists allowing non-invasive access to SR in CA1 of the dorsal hippocampus for in vivo imaging with subcellular resolution in adult mice. Neuronal activity as measured by a *cfos*-staining upon contextual fear conditioning was unchanged comparing mice with or without a hippocampal window (*Gu et al., 2014*). We have shown that microgliosis and astrogliosis decreased to contralateral hemisphere levels 3 weeks after hippocampal window implantation. However, we cannot exclude a longer-lasting inflammatory molecular signature in microglia independent of morphological changes. Another potential caveat is the use of CNO, which can have off-target effects via conversion to clozapine (*Manvich et al., 2018*; *Martinez et al., 2019*). Here, our diverse virus combinations in distinct hippocampal sub-regions led to different results, arguing against a side effect of CNO in our experiments. Cre-expression via AAVs in cell cultures was neurotoxic during development (*Forni et al., 2006*) and in dopaminergic neurons in adult mice (*Erben et al., 2022*; *Rezai Amin et al., 2019*). We cannot exclude an influence of AAV-Cre in our study, although finding opposite effects with inhibiting and activating DREADDs and the same AAV-Cre argue against it.

In summary, our study supports the hypothesis of microglia-mediated spine formation and elimination in the hippocampus of adult mice. Furthermore, our data demonstrate a link between microglial motility and neuronal activity. Microglial motility might be partially regulated via synaptically released glutamate. This has implications for in vivo studies of microglia under anesthesia. Moreover, we propose a link between microglia-spine interaction and structural plasticity of dendritic spines in the hippocampus of adult mice. A deeper understanding of the function and underlying mechanisms of microglia-synapse interactions has potential to uncover novel therapeutic strategies for many neurological and psychiatric diseases.

## Methods

### Animals

Mice were housed in the animal facility of the German Center for Neurodegenerative Diseases. They were group-housed and separated by gender with a day/night cycle of 12 hr. Water and food was accessible ad libidum. All procedures were performed in accordance with an animal protocol approved by the DZNE and the government of North-Rhine-Westphalia. CX3CR1-GFP knock-in mice carry the international name: B6.129P-CX3CR1tm1Litt/J (Stock: 5582) Jackson Laboratory. The mouse line expresses GFP in monocytes, dendritic cells, NK cells, and microglia in the brain. The *Gfp* gene is introduced as a knock-in into the allele of the *Cx3cr1* gene (chemokine C-X-C motif receptor 1). These mice were first described by *Jung et al., 2000*. YFP-H mice (B6.Cg-TgN(Thy1-YFP-H)2Jrs, Stock: 3782, Jackson Laboratory) express YFP, a spectral variant of GFP. The protein is expressed in pyramidal cells of the hippocampus and lamina V, *stratum pyramiale internum* of the cortex. Moreover, YFP expression is found on the surface of sensory neurons in dorsal root ganglia, neurons in retinal ganglia, and mossy fibers in the cerebellum. This mouse line was first described by *Feng et al., 2000*. Littermates of both sexes were randomly assigned to groups and derived from a heterozygous crossing of CX3CR1-GFP and Thy1-YFP-H. They were used for experiments at an age of 7–10 months. Notably, heterozygous mice are haploinsufficient for the fractalkine receptor (CX3CR1).

### Hippocampal window implantation

To access the dorsal region of the hippocampus, mice were unilaterally implanted with a hippocampal window as described before (*Gu et al., 2014*). Surgical tolerance and anesthesia were established with an i.p. injection of ketamine/xylazine (0.13/0.01 mg/g body weight). To prevent edema and excessive inflammation of brain tissue, dexamethasone was administered (0.2 mg/kg s.c.). For analgesia, buprenorphine (0.05 mg/kg s.c.) was injected shortly before the surgery. After surgery, the analgesic was applied three times daily for 3 consecutive days to prevent post-interventional pain and related stress reactions. Surgeries were performed 4–6 weeks prior to the first imaging session in order to allow the animal to recover from the procedure and to let the reactive gliosis and inflammation subside. This and all procedures described were in accordance with the intern regulations of the

DZNE and animal experimental protocols approved by the government of North Rhine Westphalia (84-02.04.2017.A098).

## In vivo imaging in anesthetized and awake mice

Mice were anesthetized using 5% isoflurane, which was subsequently reduced to 1% after the righting reflex had subsided. The mouse was fixed to a custom-made head-holder and mounted under the microscope on a heating plate (37°C) to keep the body temperature at optimum. Images were acquired with a Nikon 16× water immersion objective (NA 0.8) with a working distance of 3 mm. For awake imaging, we used a spinning disk made of styrofoam (r=10 cm, h=7 cm), or a custom-made linear treadmill, which allowed for imaging of running, head-fixed mice that control the speed of the spinning disk, or the treadmill independently. Prior to imaging, animals were habituated to the experimenter on 5 consecutive days. Next, mice were placed on the treadmill (same as during imaging sessions) and head-fixed for approximately 1 hr for 3–5 days. Following this, time-lapse images were acquired at $100 \times 100$ µm² in x, y-direction with a pixel size of 0.088 µm/pixel. The z-steps were 1 µm spaced for anesthetized recordings and spanned 100 µm.

The acquired z-stacks started in the CA1 pyramidal layer spanning 100 µm deep into SR. To allow for the correction of motion artifacts, oversampling in z was performed. Z-step sizes were set to 0.2 µm/step achieving an oversampling of 5× per micrometer. This allowed to manually remove any distorted or out of focus frames by carefully scrolling through each z-stack individually. Due to this oversampling and subsequent image removal, stable maximum intensity projections could be generated similar to anesthetized conditions. Time-lapse images were acquired every 5 min for a period of 45 min. Image acquisition was carried out on an upright TrimScope II (LaVision Biotech) equipped with a Ti:Sa laser (Cameleon Ultra II, Coherent) that was tuned to 920 nm to allow for simultaneous excitation of GFP and YFP with a maximum output power of 50 mW to prevent photo damage. GFP fluorescence was acquired with a 480/40 BP filterset. YFP was separated from GFP by a 510 LP filter and detected using a 535/30 BP filterset.

## TTX application

A 50 µM TTX solution in NaCl (0.9%) was prepared and stored at –20°C. In order to topically apply TTX on the hippocampus, the hippocampal window was removed under anesthesia. Subsequently the TTX solution was administered with a pipette onto the alveus and incubated for 30 min. After the incubation, a new, sterile hippocampal window was implanted. The anesthetized mouse was head-fixed under the microscope and imaging started after the narcotic had worn off.

## AAV injections

AAV injections for DREADD delivery were performed 1 week prior to the implantation of the hippocampal window. Mice were anesthetized as described before with ketamine/xylazine. The bregma was carefully exposed by removing parts of the skin with an incision over the skull midline. Small holes (∅300 µm) were drilled relative to the bregma (see positions below) and the dura was opened. Mice received unilateral stereotactic injections of an AAV2-hsyn-DIO-hM4D(Gi)-mCherry or AAV2-hsyn-DIO-hM3(Gq)-mCherry (gift from Bryan Roth, Addgene plasmid #50474) and an AAV9-CamKII-Cre (pENN.AAV.CamKII 0.4.Cre.SV40 was a gift from James M Wilson (Addgene viral prep # 105558-AAV9)) into the CA1 and CA3 region or the CA3 region alone of the right dorsal hippocampus. Coordinates for CA1 injections relative to bregma: AP –1.90 mm ML +1.50 mm DV –1.10 mm. Injections into CA3: AP –2.00 mm ML +2.50 mm DV –2.08 mm. The needle was left at the sites of injections for 10 min in total during which 0.5 µL of virus were administered at a speed of 0.1 µL/min allowing the virus to diffuse into the tissue. The skin over the skull was reattached by suturing and mice were given 1 week to recover before surgeries were performed. Analgesic regimen was performed with three times daily application of temgesic (0.05 mg/kg body weight) for 3 consecutive days.

## Application of DREADDs to modulate neuronal activity

For silencing or activating neuronal activity in the hippocampus, mice received unilateral stereotactic injections of an AAV2-hsyn-DIO-hM4D(Gi)-mCherry or AAV2-hsyn-DIO-hM3(Gq)-mCHerry together with a CamKII-Cre AAV (pENN.AAV.CamKII 0.4.Cre.SV40 was a gift from James M Wilson [Addgene viral prep # 105558-AAV9]) into the CA1 and CA3 region of the right dorsal hippocampus 1 week

prior to window implantations. In order to activate DREADDs, mice were injected intraperitoneally with 3 µg/g body weight CNO dissolved in 1% dimethyl sulfoxide (DMSO) (*López et al., 2016*) 40 min prior to the imaging session. As a control, the same mice received the solvent (1% DMSO) as vehicle injections only.

## Histology

The viral expression in the injected mice was immunohistologically validated. Mice were transcardially perfused with PBS, pH 7.4. The brains were removed and fixed overnight in 4% PFA. One hundred µm of free-floating slices were permeabilized in 0.5% Triton X-100 for 1 hr. Subsequently, slices were incubated with a mCherry antibody (1:10,000, rat serum, Thermo Fisher Scientific, M11217) in a blocking reagent (4% normal goat serum, 0.4% Triton 1%, and 4% BSA in PBS) over night at room temperature. After washing the samples three times with PBS, a secondary antibody was administered (Alexa Fluor 594 goat anti-rat, 1:400, Invitrogen, A11007) in 5% normal goat serum/BSA and incubated for 2 hr at room temperature. During the last 30 min of incubation, Nissl staining was carried out (Invitrogen, N21473, 1:200). Afterward slices were washed three times with PBS, mounted with Dako Mounting Medium, and covered with a glass cover-slip. Alternatively, a NeuN staining was performed to visualize neurons in the hippocampus using a NeuN primary antibody (1:1000, MABN140, rabbit, Sigma-Aldrich, secondary Alexa Fluor 405 goat anti-rabbit, A31556, 1:400on).

## Confocal microscopy

For confocal imaging a Zeiss LSM700 microscope was used in combination with either a 20× air objective (NA 0.8) or a 10× air objective (NA 0.3). Nissl fluorescence was acquired with a 435/455 BP filterset and excitation filter at 405 nm. Alexa Fluor 594 was excited at 555 nm and detected with a long-pass filter (LP 560 nm). The pinhole was set to 1 airy unit. Additionally, we performed semi-automated imaging using Axio Scan.Z1 (ZEISS) to visualize transfection rates of DREADDs in neurons (Plan.Apochromat 20×/0.8; $0.325 \times 0.325 \times 5\ \mu m^3$).

## Microglial motility analysis

To separate GFP and YFP, linear unmixing was performed with the unmixing tool in ZEN2010 (ZEISS). Subsequently, z-stacks were median filtered. Individual microglia cells were identified by scrolling through the stacks at each time point and then cropped using ImageJ. Z-stacks spanning around 25 µm in depth were extracted from the original z-stack. The oversampling in z-direction allowed removing distorted pictures from the z-stack by manually scrolling through the stack and deleting individual slices. The stacks were registered by applying the ImageJ 'StackReg' plugin (*Thévenaz et al., 1998*). Finally, a maximum intensity projection was carried out, resulting in a 2D projection of the cell and its branches. All recorded time points were merged into one stack and subsequently aligned using the StackReg plugin again. The time points were pseudocolored in red and green resulting in a picture in which red areas account for lost microglia branches, green for newly gained branches, and yellow pixels resembling stable parts. The turnover rate (TOR) of individual microglia processes was calculated as the number (absolute pixel value) of lost, $N_{lost}$ (red), and newly gained, $N_{gained}$ (green) pixels divided by the sum of all pixels within a determined region of interest (ROI).

$$TOR = \frac{(N_{gained} + N_{lost})}{(N_{gained} + N_{lost} + N_{stable})}$$

The ROI around a microglial cell was manually drawn in ZEN2010 according to the branching pattern displayed by individual cells. To specifically measure process turnover, we subtracted all pixels representing the cell body for each measurement consistently. Analysis was carried out blind to the experimenter condition with one exception: We could not blind the comparison of awake vs. anesthetized recordings, because the higher motion was visible in the oversampled raw data and the same person did the motion correction and motility analysis.

## Analysis of Ca²⁺ events

ROIs of axonal segments were manually drawn. For each ROI, the ΔF/F was calculated by subtracting the baseline fluorescence F0 from the signal and dividing the value by F0:

$$\Delta F/F = \frac{(F - F0)}{F0}$$

F0 was defined as the mean of the smallest 20% of all values of a time series. In order to derive an approximation of the neuronal activity from the $Ca^{2+}$-transient frequency, we inferred the underlying spiking activity from the $\Delta F/F$ traces. For spike inference, the constrained foopsi algorithm from the freely available CalmAn toolbox (*Giovannucci et al., 2019*) was used.

### Dendritic spine counting, clusters, and microglial contacts

Protrusions emerging laterally from a dendrite with a threshold of 0.4 µm in size were counted as spines independent of their individual shape. In each animal four to six radial oblique dendrites were chosen randomly for further analysis with a length spanning 20–60 µm. Spines were identified by manually scrolling through the z-stack of subsequent time points. If a spine fell below the threshold of 0.4 µm it was counted as lost. Newly formed protrusion larger than 0.4 µm were counted as new spines. We counted 4455 spines in anesthetized mice and 5365 spines in awake mice. The spine density in vivo was determined similarly as described before (*Fuhrmann et al., 2007*; *Gu et al., 2014*; *Holtmaat et al., 2005*). The gained and lost fraction ($F_{gained}$ and $F_{lost}$) of spines was calculated by dividing the number of gained spines ($N_{gained}$) and the number of lost spines ($N_{lost}$) by the number of present spines ($N_{present}$; $F_{gained} = 100 * N_{gained}/N_{present}$; $F_{lost} = 100 * N_{lost}/N_{present}$). The general turnover of dendritic spines was calculated as $(N_{lost} + N_{gained})/2*(N_{stable} + N_{lost} + N_{gained})$. For the visualization of putative microglia contacts with dendritic spines, the GFP channel (microglia) and YFP channel (dendrites/spines) were merged. Putative physical contacts of microglial processes and dendritic spines were counted by manually scrolling through the z-stack. If fluorescence of both channels was colocalized this was counted as a contact. For this the fluorescent signals had to be not more than ≤0.4 µm away from each other in the same focal plane and detectable in ≥2 focal planes. Each individual spine was checked for microglial contacts. For every time point during imaging sessions (z-stacks were spaced by 5 min each), putative contacts were analyzed. Number of analyzed spines are provided in *Table 1*. For our analyses, we included all spines visible along the dendritic segment. If a spine was newly formed (spine gain), the dendritic point from which the new spine emerged was measured, which then was subsequently used to measure microglial contacts at the dendritic shaft at d0 with the same parameters mentioned above used for spine contacts. To measure baseline contact rates at the dendritic shaft, each dendrite was subdivided into 1 µm segments. Randomly, five of these segments were tested for microglia interaction (*Urbaniak and Plous, 2013*). For exemplary pictures in the figures, microscopy images were median-filtered and z-projected using ImageJ.

Previous studies already showed preferential turnover of dendritic spines in spatial clusters along dendrites of excitatory cells in the somatosensory, motor, visual, and retrosplenial cortex (*Chen et al., 2012*; *Frank et al., 2018*; *Fu et al., 2012*). Taking into consideration that by means of two-photon microscopy the detectable spine densities of apical pyramidal neurons in the hippocampus ($\sim 1.1\ \mu m^{-1}$) (*Gu et al., 2014*) exceed those found in the cortex ($\sim 0.4\ \mu m^{-1}$) (*Holtmaat et al., 2005*; *Keck et al., 2008*; *Knott et al., 2006*) and is obviously still underestimated (*Pfeiffer et al., 2018*), we applied a stricter definition of spatial clustering than described before. Chen et al. showed spatial turnover in clusters of 10 µm on dendrites in the neocortex (*Chen et al., 2012*). Modified from this study we defined clusters as events of synaptic turnover happening within a maximum range of 4 µm to each other, accounting for the higher spine densities in the hippocampus. We defined three different classes of clustered events: lost, balanced, and gained (*Figure 3—figure supplement 1a*). If two or more spines were either gained or lost in proximity of less than 4 µm, they were classified as a 'gained' or 'lost' clustered event. If one spine was lost and another one gained within the 4 µm distance, the event was classified 'balanced'.

### Statistics

Quantifications, statistical analysis, and graph preparation were carried out using GraphPad Prism 5 and 7 (GraphPad Software Inc, La Jolla,

**Table 1.** Mouse number and gender used in the experiments.

| Mouse number Figure 4 | #Spines analyzed for contacts, anesthesia | #Spines analyzed for contacts, awake |
|---|---|---|
| 1. | 157 | 152 |
| 2. | 133 | 110 |
| 3. | 125 | 125 |
| 4. | 130 | 145 |

**Table 2.** Number of spines analyzed for contacts.

| Figure | Condition | Total number of mice | Number of male mice | Number of female mice |
|---|---|---|---|---|
| *Figure 1d* | Anesthesia | 6 | 4 | 2 |
| *Figure 1d* | Awake | 6 | 4 | 2 |
| *Figure 1d* | TTX | 4 | 3 | 1 |
| *Figure 1, Figure 1—figure supplement 1e* | $Ca^{2+}$, CA1, An vs. Aw | 3 | 3 | 0 |
| *Figure 2k* | CA3, hM4D(Gi) | 4 | 2 | 2 |
| *Figure 2l* | CA3, hM3D(Gq) | 4 | 1 | 3 |
| *Figure 2, Figure 2—figure supplement 1g* | CA1 + CA3, hM4D(Gi) | 5 | 2 | 3 |
| *Figure 2, Figure 2—figure supplement 2g* | CA3, $Ca2^+$, hM3D(Gq) | 3 | 0 | 3 |
| *Figure 3c–f* | Structural plasticity, An vs. Aw | 4 | 2 | 2 |
| *Figure 3, Figure 3—figure supplement 1* | Cluster | 4 | 2 | 2 |
| *Figure 4i–l* | Microglial contact rates | 4 | 2 | 2 |

CA, USA). Mice were assigned to experimental groups in an age and sex balanced manner (*Table 2*). Initial group sizes were calculated using G*Power (Version 3.1, University Düsseldorf, http://www.gpower.hhu.de/). Here, the primary statistical analysis method was the t-test based on the error of the first kind (0.05), second kind (0.2), biologically relevant difference (0.3) and effect size (1.5: Cohen's d). To test for normal distribution of data, D'Agostino and Pearson omnibus normality test was used for sample sizes of n>6 and the Shapiro-Wilk normality test for n<6. Statistical significance for groups of two normally distributed data sets paired or unpaired two-tailed Student's t-tests were applied. If no normal distribution was evident, Mann-Whitney test for groups of two was used. One-way ANOVA with Tukey's or Bonferroni's multiple comparison test were performed on data sets larger than two, if normally distributed. For comparison of more than two not normally distributed data sets the Kruskal-Wallis test was performed with Dunn's correction for multiple comparisons. If not indicated differently, data are represented as mean ± SEM. Figures were prepared with Illustrator CS5 Version 15.0.1 (Adobe).

## Material availability

All generated materials or mouse lines will be made available upon request.

## Material and correspondence

Materials are available on request. All correspondence should be directed to Martin Fuhrmann.

## Acknowledgements

This work was supported by the ERC-CoG MicroSynCom 865618, the DZNE, grants from the Deutsche Forschungsgemeinschaft (SFB 1089C01, B06), ERA-NET grants MicroSynDep, Microshiz. We thank P Thevenaz and Erik Meijering for the development of the ImageJ plugins 'stackreg' and 'TurboReg'. We acknowledge Bryan Roth for depositing the Addgene plasmid #44362. pENN.AAV.CamKII 0.4.Cre.SV40 was a gift from James M Wilson (Addgene viral prep # 105558-AAV9). We thank the animal and light microscopy facility of the DZNE for constant support. We thank Emily Handley, Andrew Boyce for careful reading of the manuscript, and Eleonora Ambrad Giovannetti for feedback on illustrations.

## Additional information

### Funding

| Funder | Grant reference number | Author |
|---|---|---|
| European Research Council | ERC-CoG MicroSynCom 865618 | Manuel Mittag Martin Fuhrmann |
| Deutsche Forschungsgemeinschaft | SFB1089, C01 | Martin Fuhrmann Stefanie Poll |
| Deutsche Forschungsgemeinschaft | SFB1089, B06 | Julia Steffen Martin Fuhrmann |
| Deutsche Forschungsgemeinschaft | SPP2395 | Martin Fuhrmann |
| ERA-NET NEURON | MicroSynDep | Felix Christopher Nebeling Martin Fuhrmann |
| ERA-NET NEURON | MicroSchiz | Felix Christopher Nebeling Stefanie Poll Martin Fuhrmann |

The funders had no role in study design, data collection and interpretation, or the decision to submit the work for publication.

### Author contributions

Felix Christopher Nebeling, Data curation, Formal analysis, Validation, Investigation, Visualization, Methodology, Writing - original draft, Writing - review and editing; Stefanie Poll, Resources, Investigation, Visualization; Lena Christine Justus, Investigation, Methodology; Julia Steffen, Resources, Investigation, Methodology; Kevin Keppler, Resources, Methodology; Manuel Mittag, Data curation, Software, Formal analysis, Validation, Investigation, Visualization, Methodology; Martin Fuhrmann, Conceptualization, Resources, Supervision, Funding acquisition, Validation, Writing - original draft, Project administration, Writing - review and editing

### Author ORCIDs

Stefanie Poll (iD) http://orcid.org/0000-0001-5301-2791
Martin Fuhrmann (iD) http://orcid.org/0000-0001-7672-2913

### Ethics

This and all procedures described were in accordance with the intern regulations of the DZNE and animal experimental protocols approved by the government of North Rhine Westphalia (84-02.04.2017.A098).

### Decision letter and Author response

Decision letter https://doi.org/10.7554/eLife.83176.sa1
Author response https://doi.org/10.7554/eLife.83176.sa2

## Additional files

### Supplementary files

- MDAR checklist

### Data availability

Data related to the manuscript are available according to the FAIR principles via Dryad.

The following dataset was generated:

| Author(s) | Year | Dataset title | Dataset URL | Database and Identifier |
| --- | --- | --- | --- | --- |
| Furhmann M | 2023 | Data from: Microglial motility is modulated by neuronal activity and correlates with dendritic spine plasticity in the hippocampus of awake mice | https://doi.org/10.5061/dryad.63xsj3v68 | Dryad Digital Repository, 10.5061/dryad.63xsj3v68 |

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
