## [Editor Report]

This work provides insights into the use of anesthetics in measuring cellular dynamics, regional differences in plasticity, and neuronal activity regulation of microglia dynamics. It is an important contribution to understanding hippocampal microglia plasticity in adulthood.

---

## [Decision Letter]

**Decision letter after peer review:**

[Editors’ note: the authors submitted for reconsideration following the decision after peer review. What follows is the decision letter after the first round of review.]

Thank you for submitting your work entitled "Microglia motility depends on neuronal activity and promotes structural plasticity in the hippocampus" for consideration by *eLife*. Your article has been reviewed by 3 peer reviewers, one of whom is a member of our Board of Reviewing Editors, and the evaluation has been overseen by a Senior Editor. The reviewers have opted to remain anonymous.

Our decision has been reached after consultation between the reviewers. Based on these discussions and the individual reviews below, we regret to inform you that your work will not be considered further for publication in *eLife*.

All reviewers agreed that the authors attempted to address a key question in the field of microglia motility and its association with the formation and elimination of spines in hippocampal pyramidal cells. The experiments performed by the authors are highly challenging and the reviewers were impressed by the author's technical ability to get that far in microglia imaging. Despite the enthusiasm for the work, the reviewers agreed on several major concerns that question the novelty of the influence of anesthetics on microglia motility. They were further concerned about the strength of data showing that neuronal activity influences microglia motility and, finally, they were not convinced on the effects of isoflurane on microglia motility knowing that isoflurane blocks THIK-1, which is essential for basal microglial process motility. The reviewer's therefore strongly recommended to use DREADD-mediated silencing instead isoflurane or TTX. The reviewer's had also technical concerns. The removal of the cranial window for TTX application raised concerns regarding the complications of the surgery. Moreover, the specificity of high concentrations of TTX were questioned. Finally, microglia-spine interactions may indeed, as stated by the authors, promote structural plasticity both for the individual spines, and for the dendritic segments contacted. However, the evidence for this is only correlative. A demonstration of a requirement for microglial contacts for such neuronal plasticity is not provided in the study. Therefore, the authors should modify their conclusions accordingly. Due to the time constrains of the revision phase (2 months) we propose that the authors should take their time to improve the manuscript as they see fit, based on these recommendations. It would be possible to send it back to *eLife* in the future as a new submission, should the authors choose to go that route. The individual reviews are included in their entirety below.

*Reviewer #1:*

By applying 2-Photon imaging of microglia and dendritic spines of CA1 Pyramidal cells dendrites in stratum radiatum, the authors show that (1) microglia mobility is higher in non-anaesthetized to anaesthetizes mice in vivo. (2) The motility depended on neuronal activity as shown by TTX application. (3) Microglia motility was associated with the formation of new spines and the elimination of existing spines. Stable spines were less frequently contacted by microglia. The authors therefore assume that microglia motility is associated with synaptic rewiring and neuronal connectivity.

1. Although the work is interesting the statement that microglia motility is associated with synapse formation and rewiring is bit too far stretched because no data on synapse labelling were provided.

2. Imaging micro glia extensions is under non-anaesthetized conditions a challenge due to motion artefacts and the required motion correction. Please explain in detail how motion correction was performed.

3. The DREADD experiments are interesting but evidence for changes in neuronal activity using the technique is required.

4. Although motility of microglia is an interesting topic, the question remains under which particular conditions the interaction between microglia and dendrite may support the formation of new spines and what conditions need to be met to induce a loss of spines. Since motility is dependent on neuronal activity the question emerges whether extracellular accumulation of glutamate or GABA, released from nearby synapses, may induce one or the other effect in spine dynamics.

*Reviewer #2:*

This manuscript by Nebeling et al. is well written and it set out to address in the most direct and relevant way some key questions in the field:

1. Do microglia contribute to neuronal structural plasticity in the adult brain, and if so in what way?

It appears that microglia-spine interactions may promote structural plasticity both for the individual spines, and for the dendritic segments contacted. However, the evidence for this is only correlative, and as such it only implies a functional role; a demonstration of a requirement for microglial contacts for such neuronal plasticity is not provided, so the authors should modify their conclusions accordingly.

2. Are microglial process dynamics driven by neuronal activity?

Though some prior studies in different model systems have implied this was the case, this is indeed the first direct demonstration that this is indeed the case for the adult mammalian brain, and in particular in the hippocampus, a highly plastic and relevant brain region. However, the presentation of these important results is really limited to a couple bar graphs, and no original imaging data is provided to convince the reader of the strength of these important in vivo imaging observations. In addition, the removal of the hippocampal window in order to apply TTX locally and then replace the window and image through it within the same day raises concerns regarding the complications of the surgery, additional injury, bleeding etc. Proper sham procedures and evaluation of microglial responses after only vehicle solution was applied and window was replaced would be essential for validating this experimentally challenging protocol.

3. What is the functional significance of these interactions between microglial processes and dendritic spines for the neuronal network?

The in vivo imaging approach the authors used is ideal for providing such answers and the authors attempt to provide several important answers in this way. The main finding is that anesthetics affect not only spine stability in the hippocampus (it was previously shown in the cortex), but also microglia interactions with them. The finding that anesthesia reduces microglia process motility is also not novel overall, this study only confirms it also happens in the hippocampus.

Clearly, the manuscript lacks a molecular mechanism that might explain how microglia may affect neuronal plasticity or how neuronal activity can regulate microglial process motility. Understandably, these are big questions that many in the field are trying to address with very little success to date. This study although descriptive is still very important as it offers novel insight and the first direct in vivo characterization of microglia spine interactions in the adult brain, and does so using elegant in vivo manipulations and challenging experiments. But in the absence of such mechanisms perhaps the authors should at least discuss how their data fit with proposed molecular mechanisms that have been explored in different settings such as in development and offer their perspective of those mechanisms possible relevance in the adult hippocampus.

The following points also need to be addressed:

There is no demonstration that the DREADD affected neurons are the ones being imaged. Sup. Figure S4 is not properly called out in the results (S2 is mentioned instead). Still, in S4 the overlay of mCherry fluorescence with Nissl poorly demonstrates the success of the injections, a NeuN immunostaining would have been more effective. The labeling of the figure overlaps with the image, which should not happen.

Most importantly, Figure 2 does not demonstrate any representative data to evaluate how the authors appreciated the result that was quantified in the bar graph. Representative images for each condition should be presented along with the quantification. It would be great if the authors could demonstrate that they indeed imaged M4+ neurons.

Figure 3: The low power images in b and e are very small and hard to appreciate. Orange arrows are described in the legend but absent in the figure. In d and g, no information is provided about when (at which time point) these timelapse images were acquired. It is also hard to appreciate what is considered to be a lost and what a gained spine in these images, as the structural changes are tiny. All the labeling inside these pictures is tiny and very hard to read at the full figure size.

In figure S2 the removal of background signal is not described anywhere in the methods. Some of the protrusions are very faint and hard to tell if they are more than background noise, especially the top red and green ones. Also the bottom green one seems to be preexisting on d0, that only grew in size by d2. The authors should pick better representative data that clearly show what the authors claim.

In Supplementary movie legends, there is no information about the z-depth of these stacks. If the z-depths are not extensive like it would appear, then more frequent sampling data if available would be more effective in showing the microglia process motility. This would be more informative, but not necessary if such data are not available.

The results regarding the clustering of spines on page 6 are poorly described.

The data showed an association of microglia contacts of dendritic spines with spine plasticity not stability as is stated in the discussion

In Methods: information that is missing and should be provided:

1. The specific lenses used for imaging the hippocampus in vivo.

2. The depth of z-stacks (how much oversampling).

3. Temgesic is not an active ingredient.

4. Doses should be provided in an equivalent way throughout (for example mg/g of BW).

5. The total length of dendrites sampled is not well explained or justified. If a few small segments per mouse are selected, then the selection criteria need to be explained. 20μm is too short and if they are selected based on how obvious changes on them appear, that can introduce sampling bias. The authors should address how they avoid introducing such biases in their analyses.

6. The gained and lost fraction is a simple percentage calculation and it should just be called that, no formula is necessary. On the contrary, the calculation of the "general turnover of dendritic spines" is not explained; what does this formula actually calculate?

7. The explanation of the dendritic shaft versus contact base sampling is very confusing and the reader cannot understand how the analysis was performed.

8. The TOR needs to be better explained in the methods: what does "the number of red and green pixels" mean? are red and green pixels added and then divided by total number of red green and yellow pixels? if this is done in "specified ROIs how are those picked? why is this not done for the entire process tree of each cell? Are cell body pixels always subtracted or only sometimes?

*Reviewer #3:*

In this report, Fuhrmann and colleagues sought to address the role of microglia in hippocampal synaptic connectivity. The research question is unambiguously important as persuasive data could locate microglia in a crucial position for hippocampal connectivity, and provide evidence favoring their involvement in higher cognitive functions under physiological conditions. The paper's title "Microglia motility depends on neuronal activity and promotes structural plasticity in the hippocampus" implies that this standard has been met. There are features of innovation in the research study, including use of two-photon imaging to visualize hippocampal microglial process motility and relation of processes to neuronal elements. Collection of these data was achieved by removing overlying cortical elements, an approach which hasn't been widely, if ever, applied to this research question. Additionally, the authors used DREADDs to silence neurons in the experiments shown in Figure 2, providing encouraging if preliminary data supporting their hypothesis.

Unfortunately, most of the data in this study come from multiply confounded experiments:

1. Isoflurane effect on microglial motility per se, independent of effect on neuronal activity: Isoflurane also blocks microglial THIK-1 which is essential for basal microglial process motility (Attwell Neuron 2018). How does one overcome the effect of isoflurane on microglial motility to reach the conclusions articulated in the text? Can the experiments be done using DREADD-mediated silencing instead?

2. TTX effect on Na^+^ channels of microglia: How specific is TTX at this high concentration? Any effect on microglial Na^+^ channels?

In that regard, two papers from a prominent group of NaV channel researchers, show effects of TTX on microglia at submicromolar concentrations:

"Contribution of sodium channels to lamellipodial protrusion and Rac1 and ERK1/2 activation in ATP-stimulated microglia." Persson AK…Waxman SG, Black JA. Glia. 2014 Dec;62(12):2080-95. doi: 10.1002/glia.22728. Epub 2014 Jul 18. PMID: 25043721

"Sodium channel activity modulates multiple functions in microglia." Black JA, Liu S, Waxman SG. Glia. 2009 Aug 1;57(10):1072-81. doi: 10.1002/glia.20830.

3. Effect of Cx3cr1 haploinsufficiency: need to specify that microglial haploinsufficiency for Cx3cr1 precludes generalizing observations to WT microglia.

All these experiments (Figures 1, 3) should be deleted in favor of those shown in Figure 2 using DREADDs to silence neurons and WT rather than Cx3cr1-haploinsufficient microglia. The conclusions are unchanged.

[Editors’ note: further revisions were suggested prior to acceptance, as described below.]

Thank you for resubmitting your work entitled "Microglial motility is modulated by neuronal activity and correlates with dendritic spine plasticity in the hippocampus of awake mice" for further consideration by *eLife*. Your revised article has been evaluated by Laura Colgin (Senior Editor).

The manuscript has been improved but there are some remaining issues that need to be addressed, as outlined below:

The general recommendations to authors are to: (1) better discuss the limitations of their techniques, (2) better discuss potential caveats of their study, (3) put their work in the context of more of the existing literature, (4) re-do some of the statistics, and (5) include histology to show the extent of damage that is done to the tissue after implantation and imaging, if available; alternatively, caveats associated with a lack of histology could be mentioned. Other minor text revisions are recommended to provide missing details and to improve clarity. Please refer to the individual reviews below for details.

*Reviewer #1 (Recommendations for the authors):*

The authors have improved the manuscript and have added new data and analyses as well as raw data, and methodological details according to the reviewers' suggestions. They have also toned down the conclusions from their study to refrain from causality statements and underscore correlative relationships. They discuss their results in the context of prior studies and highlight similarities and possible explanations for differences. The study is of value to the field in its current state.

*Reviewer #2 (Recommendations for the authors):*

Nebeling et al. seek to: (1) characterize microglial motility under healthy conditions and in the absence of anesthesia in the hippocampus of awake, adult mice; (2) examine to what extent baseline surveillance of microglia is dependent on neuronal activity; and (3) address the question of whether microglia play a role in dendritic spine formation and elimination in the hippocampus during adulthood. To do this, the authors used appropriate experimental approaches – all experiments were conducted using hippocampal cranial window preparations (as described in previous work) and two-photon microscopy to image the same mice under different experimental conditions. They also used the DREADD system to modulate input from CA3 to CA1 to determine how changes in neuronal activity regulate microglia dynamics. Lastly, they monitored both microglia and dendritic spines and measured rates of spine turnover, microglia-dendritic spine contact, and microglia-dendritic shaft contact. The authors found that CA3 to CA1 input is sufficient to modulate microglial process motility, more dendritic spines emerged in awake compared to anesthetized mice, and microglial contact rates of individual dendritic spines and dendrites were associated with the stability, removal, or emergence of dendritic spines. This work provides impressive new information on the use of anesthetics in measuring cellular dynamics, regional differences in plasticity, and neuronal activity regulation of microglia dynamics. Overall, it is an important contribution to understanding hippocampal microglia plasticity in adulthood.

1) It is interesting that there is a significant increase in dendritic spines density and significantly higher fractions of gained to lost spines in awake mice (Figure 3). This suggests that over time, more and more spines are being formed throughout adulthood in the hippocampus rather than being maintained through a balance in formation and loss of spines over time. The authors explain this in one sentence on line 360: "We interpret the increased spine density after awake imaging most likely being a response to an enriched environment, which has been shown to induce a dendritic spine increase in the hippocampus (Rampon et al., 2000) and the cortex (Jung and Herms, 2014)". While it is certainly possible that 1 hour of being on a treadmill (is this what the authors mean by enriched environment?) could temporarily effect spine gain, it seems to me much more likely that the authors are measuring a stress-induced effect and not really comparing anesthetized to awake conditions. This appears all the more likely given that the authors did not describe any head-fixation habituation/awake training in the methods. It is standard for mice to undergo several training sessions to habituate to head restraint before collecting experimental data because head fixation is stressful (see references below). If the mice used in these studies underwent habituation, please add this to your methods. If not, this needs to be thoroughly discussed as a possible confound of the findings in the Discussion section. It would be preferable to have some measure of stress signaling (cortisol ELISA) after both imaging sessions, but I realize this would require more difficult experiments so at the very least please discuss this possibility and the caveats of the current approach. Furthermore, if the running apparatus (treadmill or foam running wheel) is not consistent between training and imaging, this could further impact your results. In any case, some of the statements made in the manuscript seem disingenuous, like that on line 217: "Here, we observed a significant increase in spine density, as well as significantly higher fractions of gained compared to lost spines in awake mice (Figure 3c-f). These results underscore the rewiring potential of hippocampal synaptic connectivity in awake mice during adulthood". And 359: "Indeed, awake mice exhibited a higher spine density after two days compared with anesthetized mice". These kinds of statements suggest that there is something inherent in awake mice that induces gain of spines as compared to anesthetized mice. Given that mice are anesthetized for a very short period of imaging over two days and the majority of spine gain occurs in the awake condition in both sets of comparisons, it seems overblown to make statements about spine turnover under anesthesia vs. wakefulness. The statements on microglia-spine contact and the outcome 2 days later are fine but please refrain from inferring remodeling in circuits based on these experiments.

a. Examples of habituation period prior to imaging with head fixation:

i. Giovannucci, A. et al. Automated gesture tracking in head-fixed mice. J. Neurosci. Methods 300, 184-195. https://doi.org/10.1016/j.jneumeth.2017.07.014 (2018).

ii. Chettih, S. N., McDougle, S. D., Ruffolo, L. I. and Medina, J. F. Adaptive timing of motor output in the mouse: the role of movement oscillations in eyelid conditioning. Front. Integr. Neurosci. 5, 72. https://doi.org/10.3389/fnint.2011.00072 (2011).

iii. Stowell, R. D., Sipe, G. O., Dawes, R. P., Batchelor, H. N., Lordy, K. A., Whitelaw, B. S., … and Majewska, A. K. (2019). Noradrenergic signaling in the wakeful state inhibits microglial surveillance and synaptic plasticity in the mouse visual cortex. Nature neuroscience, 22(11), 1782-1792.

b. Examples of how stress can influence data collected with head restraint:

i. Juczewski, K., Koussa, J. A., Kesner, A. J., Lee, J. O., and Lovinger, D. M. (2020). Stress and behavioral correlates in the head-fixed method: stress measurements, habituation dynamics, locomotion, and motor-skill learning in mice. Scientific reports, 10(1), 1-19.

2) A more detailed description of microglia contact analysis should be provided. The number of spines analyzed for contacts should be included (was this the same numbers as the turnover analysis?). Once a contacted spine was identified, were neighboring spines that were not contacted also included in the analysis? If not, when comparing contacted versus non-contacted spines, any difference in turnover could be due to the probability of contact/physical limits of the microglia rather than a property of the spine itself. It is also important to mention that these contacts are putative, and that electron microscopy would be needed to fully resolve contact.

3) It is unclear to me why analysis in Figure 1-2 uses animals as the statistical n while in Figure 3-4 dendrites are used (is this true for Figure 4 – it was not clear). At the very least, please provide supplementary figures that show the analysis averaged per animal so the reader can evaluate any animal-based trends in the data. Please list the # of spines/mouse/day imaged.

4) The authors chose to only use one main methodology for their experiments- in vivo imaging using two-photon microscopy utilizing a hippocampal cranial window preparation- which is an appropriate method to measure cellular dynamics. The experiments are technically challenging. However, the authors could do more to discuss the limitations of using their techniques and put their work in the context of the literature.

*Reviewer #3 (Recommendations for the authors):*

The authors address an important question in the field-how do microglia in the hippocampus respond to local neuronal activity? They show that microglial motility increases in response to activity. The most convincing piece of evidence is the DREADD data, which was added in response to the comments of previous reviewers. These data are novel and will improve our understanding of memory disorders and neurodegeneration, where hippocampal microglia likely play a role.

However, as previous reviewers pointed out, the experimental design is not optimal, which is not an easy problem to mitigate. The authors seem to have addressed some of the concerns by mentioning important caveats and moderating some of the claims. However, this resulted in the loss of significance of some of conclusions.

Strengths:

1. The new DREADD experiment addresses an important question in the field-how do microglia in the hippocampus respond to local neuronal activity? The authors now show causal evidence that microglial motility increases in response to activity. Their data are novel and will improve our understanding of memory disorders and neurodegeneration, where hippocampal microglia likely play a role (but see caveats below).

2. While some of the isoflurane vs. anesthetized data are perhaps not entirely novel, they still contribute to reconciling the findings from previous publications in the field.

Weaknesses:

1. The window implantation may result in increased inflammation in the hippocampus, and will certainly trigger inflammatory responses in the cortex. Moreover, it is possible that inflammation is present in some mice but not others, depending on how much tissue was removed during the implantation surgery and whether hippocampal tissue was damaged.

2. The experimental timeline is another potential concern, given the issues with inflammation discussed in the previous point. Is there a reason for the 3-week interval between anesthesia, awake and TTX experiments in Figure 1? Since the interval between imaging sessions is quite long, could it be that the time from surgery affects the inflammatory response in the 3 different conditions? Along the same lines, could the 1-week wait time between baseline and CNO experiments in Figure 2 be excessive? Were the experiments counterbalanced so that some mice received CNO in the first session, while others received it in the second session?

3. The TTX experiments are difficult to interpret due to the acute injury response from the TTX injection, which could potentially induce microglia migration in the area adjacent to the lesion.

4. Another potential confound is the AAV-mediated over-expression of Cre recombinase, which can have toxic effects.

5. I am not sure that the statistics in Figure 3 and Figure 4 are appropriate. The measurements from different dendrites are not truly independent observations since some were collected in the same mouse (i.e. the data is nested).

Suggested approach to mitigate weaknesses:

Weakness 1: The authors could discuss this caveat and quantify neuroinflammatory markers such as GFAP, microglial density and possibly microglial morphology (although from the images shown microglial do not appear to be ameboid). The non-implanted hemisphere or non-implanted tissue can be used as controls.

Weakness 3: This is already addressed in the text in response to the previous reviewers but it is nevertheless a weakness that reduces the significance of the data.

Weakness 4: Mention caveat in text

Weakness 6: It seems to me that a multilevel statistical model should be used (even more so since the number of dendrites imaged per mouse seems to be variable).

---

## [Author Response]

[Editors’ note: the authors resubmitted a revised version of the paper for consideration. What follows is the authors’ response to the first round of review.]

Reviewer #1:By applying 2-Photon imaging of microglia and dendritic spines of CA1 Pyramidal cells dendrites in stratum radiatum, the authors show that (1) microglia mobility is higher in non-anaesthetized to anaesthetizes mice in vivo. (2) The motility depended on neuronal activity as shown by TTX application. (3) Microglia motility was associated with the formation of new spines and the elimination of existing spines. Stable spines were less frequently contacted by microglia. The authors therefore assume that microglia motility is associated with synaptic rewiring and neuronal connectivity.1. Although the work is interesting the statement that microglia motility is associated with synapse formation and rewiring is bit too far stretched because no data on synapse labelling were provided.

We thank the reviewer for the valuable comment. We have modified the corresponding text passages accordingly and more precisely use “dendritic spines” instead of “synapses”. However we think monitoring dendritic spines is a good surrogate marker to draw conclusions about post-synapses as the majority (~80%, in adults) contain PSD-95 which corresponds to mature excitatory synapses (Runge *et al.*, 2021, Cane *et al.* 2014).

2. Imaging micro glia extensions is under non-anaesthetized conditions a challenge due to motion artefacts and the required motion correction. Please explain in detail how motion correction was performed.

We have incorporated a detailed section in the material and methods section. In brief, stable image quality was achieved by oversampling in the z dimension by a factor of 5x compared to imaging of microglia under anesthetized conditions. This allows for the exclusion of distorted images generated by movement of the animal.

3. The DREADD experiments are interesting but evidence for changes in neuronal activity using the technique is required.

To address this point, we have performed ca^2+^-imaging of axonal fibers (Schaffer collaterals) in *str. radiatum* of the hippocampus (new Figure 2 —figure supplement 2, Figure 2 – Video 1+2). CA3 neurons were co-transfected with a CaMKII-Cre, a loxP flanked GCaMP6m and a loxP flanked hM3D(Gq) DREADD. Mice were imaged awake on a linear treadmill upon vehicle and CNO application. We found increased axonal ca^2+^-event frequencies upon hM3D(Gq) activation via CNO administration. Our data is in line with previous experiments showing the effect of DREADD activation on neuronal activity (Alexander et. al 2009, Roth 2016). We are therefore confident that our DREADD experiments indeed change neuronal activity.

4. Although motility of microglia is an interesting topic, the question remains under which particular conditions the interaction between microglia and dendrite may support the formation of new spines and what conditions need to be met to induce a loss of spines. Since motility is dependent on neuronal activity the question emerges whether extracellular accumulation of glutamate or GABA, released from nearby synapses, may induce one or the other effect in spine dynamics.

Thank you for this valuable and important comment. In a follow up project we are currently actually investigating these mechanisms. However, we think it is out of the scope of the current manuscript to provide this mechanism. We therefore elaborate this important point in the discussion. “Potentially the microglial response to increased levels of glutamate might be due to activity-induced glutamate uptake and the subsequent release of GABA by astrocytes; this might function as a relay station between synapses and microglia, triggering a microglial response through the activation of GABAB receptors on microglia (Logiacco et al., 2021).”

Reviewer #2:This manuscript by Nebeling et al. is well written and it set out to address in the most direct and relevant way some key questions in the field:1. Do microglia contribute to neuronal structural plasticity in the adult brain, and if so in what way?It appears that microglia-spine interactions may promote structural plasticity both for the individual spines, and for the dendritic segments contacted. However, the evidence for this is only correlative, and as such it only implies a functional role; a demonstration of a requirement for microglial contacts for such neuronal plasticity is not provided, so the authors should modify their conclusions accordingly.

We thank the reviewer for the valuable comment. We modified our conclusion throughout the manuscript not claiming causality anymore, instead describing the correlation between microglia contacts and structural plasticity.

2. Are microglial process dynamics driven by neuronal activity?Though some prior studies in different model systems have implied this was the case, this is indeed the first direct demonstration that this is indeed the case for the adult mammalian brain, and in particular in the hippocampus, a highly plastic and relevant brain region. However, the presentation of these important results is really limited to a couple bar graphs, and no original imaging data is provided to convince the reader of the strength of these important in vivo imaging observations. In addition, the removal of the hippocampal window in order to apply TTX locally and then replace the window and image through it within the same day raises concerns regarding the complications of the surgery, additional injury, bleeding etc. Proper sham procedures and evaluation of microglial responses after only vehicle solution was applied and window was replaced would be essential for validating this experimentally challenging protocol.

We agree that the removal and consecutive application of TTX represents a difficult experiment that has certain drawbacks. To support the results of the pharmacologic TTX experiment, we performed a more specific chemo-genetic experiment using DREADDs to alter neuronal activity, while imaging microglia dynamics at the same time. These experiments revealed a positive correlation between neuronal activity and microglia activity in the hippocampus of adult mice. Furthermore, we provide original data (see Figure 1 – Video 1+2, and *new* Figure 2 + Figure 2 —figure supplement 1) to support our graphs.

3. What is the functional significance of these interactions between microglial processes and dendritic spines for the neuronal network?The in vivo imaging approach the authors used is ideal for providing such answers and the authors attempt to provide several important answers in this way. The main finding is that anesthetics affect not only spine stability in the hippocampus (it was previously shown in the cortex), but also microglia interactions with them. The finding that anesthesia reduces microglia process motility is also not novel overall, this study only confirms it also happens in the hippocampus.

Given that microglia have distinct region-dependent transcriptional identities that vary significantly e.g. between cortex and hippocampus (Grabert *et al.,* 2016), investigating microglial properties like their baseline motility in different brain regions can give further valuable insights into their brain region-dependent functional properties. As such our study is novel also in the light of recent findings that microglial motility in the cortex might actually be differentially regulated than in the hippocampus (Stowell *et al.*, 2019). We are convinced that our findings concerning the hippocampus are novel and are therefore of interest for the neuroscientific community.

Clearly, the manuscript lacks a molecular mechanism that might explain how microglia may affect neuronal plasticity or how neuronal activity can regulate microglial process motility. Understandably, these are big questions that many in the field are trying to address with very little success to date. This study although descriptive is still very important as it offers novel insight and the first direct in vivo characterization of microglia spine interactions in the adult brain, and does so using elegant in vivo manipulations and challenging experiments. But in the absence of such mechanisms perhaps the authors should at least discuss how their data fit with proposed molecular mechanisms that have been explored in different settings such as in development and offer their perspective of those mechanisms possible relevance in the adult hippocampus.

We fully agree that the discovery of a molecular mechanism would have been a great add to our manuscript as well as for field. To properly represent the importance of identifying a mechanism, we elaborate on this in the discussion. E.g. we discuss the probability auf neurotransmitter modulation of microglia-spine interaction.

The following points also need to be addressed:There is no demonstration that the DREADD affected neurons are the ones being imaged. Sup. Figure S4 is not properly called out in the results (S2 is mentioned instead). Still, in S4 the overlay of mCherry fluorescence with Nissl poorly demonstrates the success of the injections, a NeuN immunostaining would have been more effective. The labeling of the figure overlaps with the image, which should not happen.

We thank the reviewer for the valuable comment. We provide new histological examinations of DREADD transfected neurons and implemented the statistical analysis alongside representative images for the new conditions applied (also please see next point, Figure 2, Figure 2 —figure supplement 1).

Most importantly, Figure 2 does not demonstrate any representative data to evaluate how the authors appreciated the result that was quantified in the bar graph. Representative images for each condition should be presented along with the quantification. It would be great if the authors could demonstrate that they indeed imaged M4+ neurons.

We now included representative data in example images. Furthermore, we measured the transfection rate of neurons by DREADDs (hM4Di+hM3Dq) and show the corresponding data in Figure 2.

Figure 3: The low power images in b and e are very small and hard to appreciate. Orange arrows are described in the legend but absent in the figure. In d and g, no information is provided about when (at which time point) these timelapse images were acquired. It is also hard to appreciate what is considered to be a lost and what a gained spine in these images, as the structural changes are tiny. All the labeling inside these pictures is tiny and very hard to read at the full figure size.

We have improved figure 3 (now termed Figure 4) and implemented a better exemplary picture for the loss of a dendritic spine.

In figure S2 the removal of background signal is not described anywhere in the methods. Some of the protrusions are very faint and hard to tell if they are more than background noise, especially the top red and green ones. Also the bottom green one seems to be preexisting on d0, that only grew in size by d2. The authors should pick better representative data that clearly show what the authors claim.

We thank the reviewer for careful reading. The images were median filtered and z-projected.

This is described in Material and Methods now. We have improved Figure S2 (now termed Figure 3) and provided a better exemplary image.

In Supplementary movie legends, there is no information about the z-depth of these stacks. If the z-depths are not extensive like it would appear, then more frequent sampling data if available would be more effective in showing the microglia process motility. This would be more informative, but not necessary if such data are not available.

We have provided information of respective z-profiles for this data.

The results regarding the clustering of spines on page 6 are poorly described.The data showed an association of microglia contacts of dendritic spines with spine plasticity not stability as is stated in the discussion

We improved the description how the analysis of spine clustering was carried out.

Furthermore, we thank the reviewer for careful reading. We revised the section and

clarify that microglia contacts of dendritic spines is associated with spine plasticity.

In Methods: information that is missing and should be provided:1. The specific lenses used for imaging the hippocampus in vivo.

Nikon 16x NA0.8, WD 3.0mm, water immersion.

2. The depth of z-stacks (how much oversampling).

100 μm, every 0.2 μm 5x.

3. Temgesic is not an active ingredient.

Now buprenorphine.

4. Doses should be provided in an equivalent way throughout (for example mg/g of BW).

Done.

5. The total length of dendrites sampled is not well explained or justified. If a few small segments per mouse are selected, then the selection criteria need to be explained. 20μm is too short and if they are selected based on how obvious changes on them appear, that can introduce sampling bias. The authors should address how they avoid introducing such biases in their analyses.

Dendritic segments were randomly selected. Each dendrite had a length between 20- 60 μm. Dendritic spine density of dendrites of CA1 neurons are more than two times higher than on cortical dendrites (Gu et al. 2014, Attardo et al. 2015, Pfeiffer et al. 2018). Therefore, a segment of 20 μm length has a sufficiently high spine number.

6. The gained and lost fraction is a simple percentage calculation and it should just be called that, no formula is necessary. On the contrary, the calculation of the "general turnover of dendritic spines" is not explained; what does this formula actually calculate?

We included the calculation of spine turnover.

7. The explanation of the dendritic shaft versus contact base sampling is very confusing and the reader cannot understand how the analysis was performed.

Is better described now in the text and figure legend.

8. The TOR needs to be better explained in the methods: what does "the number of red and green pixels" mean? are red and green pixels added and then divided by total number of red green and yellow pixels? if this is done in "specified ROIs how are those picked? why is this not done for the entire process tree of each cell? Are cell body pixels always subtracted or only sometimes?

We provide a formula and precise way how it was carried out.

We have reworked all of the 8 remarks pointed out above and provide the information requested.

Reviewer #3:In this report, Fuhrmann and colleagues sought to address the role of microglia in hippocampal synaptic connectivity. The research question is unambiguously important as persuasive data could locate microglia in a crucial position for hippocampal connectivity, and provide evidence favoring their involvement in higher cognitive functions under physiological conditions. The paper's title "Microglia motility depends on neuronal activity and promotes structural plasticity in the hippocampus" implies that this standard has been met. There are features of innovation in the research study, including use of two-photon imaging to visualize hippocampal microglial process motility and relation of processes to neuronal elements. Collection of these data was achieved by removing overlying cortical elements, an approach which hasn't been widely, if ever, applied to this research question. Additionally, the authors used DREADDs to silence neurons in the experiments shown in Figure 2, providing encouraging if preliminary data supporting their hypothesis.Unfortunately, most of the data in this study come from multiply confounded experiments:1. Isoflurane effect on microglial motility per se, independent of effect on neuronal activity: Isoflurane also blocks microglial THIK-1 which is essential for basal microglial process motility (Attwell Neuron 2018). How does one overcome the effect of isoflurane on microglial motility to reach the conclusions articulated in the text? Can the experiments be done using DREADD-mediated silencing instead?

We thank the reviewer for the valuable comment. We provide further DREADDsilencing and activation of CA3 neurons additionally to the already applied DREADDsilencing included in the initial submission (See Figure 2). We are confident that these experiments support our conclusion of a relationship between microglia motility and neuronal activity in the hippocampus.

2. TTX effect on Na^+^ channels of microglia: How specific is TTX at this high concentration? Any effect on microglial Na^+^ channels?In that regard, two papers from a prominent group of NaV channel researchers, show effects of TTX on microglia at submicromolar concentrations:"Contribution of sodium channels to lamellipodial protrusion and Rac1 and ERK1/2 activation in ATP-stimulated microglia." Persson AK…Waxman SG, Black JA. Glia. 2014 Dec;62(12):2080-95. doi: 10.1002/glia.22728. Epub 2014 Jul 18. PMID: 25043721"Sodium channel activity modulates multiple functions in microglia." Black JA, Liu S, Waxman SG. Glia. 2009 Aug 1;57(10):1072-81. doi: 10.1002/glia.20830.

We agree that the usage of isoflurane and especially the topic application of TTX and the potential direct effect on microglia needs to be evaluated critically. As stated above we have improved our manuscript by adding in vivo manipulation of neuronal activity by DREADD silencing and activation experiments. However, we did not perform any experiments in the direction of Na^+^ channel manipulation on microglia. To take this important possible mechanism into account, we included it in the discussion.

3. Effect of Cx3cr1 haploinsufficiency: need to specify that microglial haploinsufficiency for Cx3cr1 precludes generalizing observations to WT microglia.

We have stated this now in the manuscript and clarified that these mice are haploin-sufficient for the fractalkine receptor. Therefore, the results are of course not generalizable to WT microglia.

All these experiments (Figures 1, 3) should be deleted in favor of those shown in Figure 2 using DREADDs to silence neurons and WT rather than Cx3cr1-haploinsufficient microglia. The conclusions are unchanged.

We understand the concern about the usage of the Cx3Cr1-GFP mouse line, but currently do not have a better model at hand in terms of haploinsufficiency. Furthermore, the findings shown in Figure 3 (now termed Figure 4) concerning the interaction of microglia with dendritic spines in the adult hippocampus is of particular interest and novel to the field. Therefore, we decided to keep figures 1 and 3 (Figure 4 respectively).

[Editors’ note: what follows is the authors’ response to the second round of review.]

Reviewer #2 (Recommendations for the authors):Nebeling et al. seek to: (1) characterize microglial motility under healthy conditions and in the absence of anesthesia in the hippocampus of awake, adult mice; (2) examine to what extent baseline surveillance of microglia is dependent on neuronal activity; and (3) address the question of whether microglia play a role in dendritic spine formation and elimination in the hippocampus during adulthood. To do this, the authors used appropriate experimental approaches – all experiments were conducted using hippocampal cranial window preparations (as described in previous work) and two-photon microscopy to image the same mice under different experimental conditions. They also used the DREADD system to modulate input from CA3 to CA1 to determine how changes in neuronal activity regulate microglia dynamics. Lastly, they monitored both microglia and dendritic spines and measured rates of spine turnover, microglia-dendritic spine contact, and microglia-dendritic shaft contact. The authors found that CA3 to CA1 input is sufficient to modulate microglial process motility, more dendritic spines emerged in awake compared to anesthetized mice, and microglial contact rates of individual dendritic spines and dendrites were associated with the stability, removal, or emergence of dendritic spines. This work provides impressive new information on the use of anesthetics in measuring cellular dynamics, regional differences in plasticity, and neuronal activity regulation of microglia dynamics. Overall, it is an important contribution to understanding hippocampal microglia plasticity in adulthood.1) It is interesting that there is a significant increase in dendritic spines density and significantly higher fractions of gained to lost spines in awake mice (Figure 3). This suggests that over time, more and more spines are being formed throughout adulthood in the hippocampus rather than being maintained through a balance in formation and loss of spines over time. The authors explain this in one sentence on line 360: "We interpret the increased spine density after awake imaging most likely being a response to an enriched environment, which has been shown to induce a dendritic spine increase in the hippocampus (Rampon et al., 2000) and the cortex (Jung and Herms, 2014)". While it is certainly possible that 1 hour of being on a treadmill (is this what the authors mean by enriched environment?) could temporarily effect spine gain, it seems to me much more likely that the authors are measuring a stress-induced effect and not really comparing anesthetized to awake conditions. This appears all the more likely given that the authors did not describe any head-fixation habituation/awake training in the methods. It is standard for mice to undergo several training sessions to habituate to head restraint before collecting experimental data because head fixation is stressful (see references below). If the mice used in these studies underwent habituation, please add this to your methods. If not, this needs to be thoroughly discussed as a possible confound of the findings in the Discussion section. It would be preferable to have some measure of stress signaling (cortisol ELISA) after both imaging sessions, but I realize this would require more difficult experiments so at the very least please discuss this possibility and the caveats of the current approach. Furthermore, if the running apparatus (treadmill or foam running wheel) is not consistent between training and imaging, this could further impact your results. In any case, some of the statements made in the manuscript seem disingenuous, like that on line 217: "Here, we observed a significant increase in spine density, as well as significantly higher fractions of gained compared to lost spines in awake mice (Figure 3c-f). These results underscore the rewiring potential of hippocampal synaptic connectivity in awake mice during adulthood". And 359: "Indeed, awake mice exhibited a higher spine density after two days compared with anesthetized mice". These kinds of statements suggest that there is something inherent in awake mice that induces gain of spines as compared to anesthetized mice. Given that mice are anesthetized for a very short period of imaging over two days and the majority of spine gain occurs in the awake condition in both sets of comparisons, it seems overblown to make statements about spine turnover under anesthesia vs. wakefulness. The statements on microglia-spine contact and the outcome 2 days later are fine but please refrain from inferring remodeling in circuits based on these experiments.a. Examples of habituation period prior to imaging with head fixation:i. Giovannucci, A. et al. Automated gesture tracking in head-fixed mice. J. Neurosci. Methods 300, 184-195. https://doi.org/10.1016/j.jneumeth.2017.07.014 (2018).ii. Chettih, S. N., McDougle, S. D., Ruffolo, L. I. and Medina, J. F. Adaptive timing of motor output in the mouse: the role of movement oscillations in eyelid conditioning. Front. Integr. Neurosci. 5, 72. https://doi.org/10.3389/fnint.2011.00072 (2011).iii. Stowell, R. D., Sipe, G. O., Dawes, R. P., Batchelor, H. N., Lordy, K. A., Whitelaw, B. S., … and Majewska, A. K. (2019). Noradrenergic signaling in the wakeful state inhibits microglial surveillance and synaptic plasticity in the mouse visual cortex. Nature neuroscience, 22(11), 1782-1792.b. Examples of how stress can influence data collected with head restraint:i. Juczewski, K., Koussa, J. A., Kesner, A. J., Lee, J. O., and Lovinger, D. M. (2020). Stress and behavioral correlates in the head-fixed method: stress measurements, habituation dynamics, locomotion, and motor-skill learning in mice. Scientific reports, 10(1), 1-19.

We thank the reviewer for this helpful comment. We agree with the reviewer that the increase in gained spines after 1h awake imaging is very likely transient. We and others have shown that under baseline conditions, spine densities remain at a steady state level. We also habituated the mice and included the description in the MatandMet. part now. The treadmill for training and experiment was the same. We are therefore confident that the spine gain effect was not stress-related. To clarify the point that the spine gain is likely transient, we revised the part of the discussion addressing this point: “Indeed, awake mice, meaning imaged under awake conditions while being on a treadmill for one hour, exhibited a higher spine density after two days compared with anesthetized mice. We suspect the increase observed in our study to be transient, as permanent increases in spine formation was not observed in previous studies (Attardo et al., 2015; Gu et al., 2014; Pfeiffer et al., 2018). Although mice were habituated, it is possible that the one hour awake imaging is similar to an enriched environment, which induced a dendritic spine increase in the hippocampus (Rampon et al., 2000) and the cortex (Jung and Herms, 2014).” (L361-368)

We also revised the sentences L217, L359 (now L219 and L361) to clarify that awake mice do not inherently have higher spine gain. Furthermore, we refrain from inferring remodeling circuits based on our experiments.

2) A more detailed description of microglia contact analysis should be provided. The number of spines analyzed for contacts should be included (was this the same numbers as the turnover analysis?). Once a contacted spine was identified, were neighboring spines that were not contacted also included in the analysis? If not, when comparing contacted versus non-contacted spines, any difference in turnover could be due to the probability of contact/physical limits of the microglia rather than a property of the spine itself. It is also important to mention that these contacts are putative, and that electron microscopy would be needed to fully resolve contact.

We thank the reviewer for this helpful comment. We now provide a more detailed description of microglia contact analysis, as well as the number of spines analyzed (see MatandMet L678-694; Table 2). The same number of spines was analyzed for contacts and turnover. All spines along a dendritic segment were taken into account, which means that neighboring spines were also included. Furthermore, we clarify in the discussion that these are putative contacts (L422).

3) It is unclear to me why analysis in Figure 1-2 uses animals as the statistical n while in Figure 3-4 dendrites are used (is this true for Figure 4 – it was not clear). At the very least, please provide supplementary figures that show the analysis averaged per animal so the reader can evaluate any animal-based trends in the data. Please list the # of spines/mouse/day imaged.

We thank the reviewer for pointing this out. We provide the statistics over mice now and changed Figure 3, 4 accordingly. The analysis over dendrites is provided as supplementary file (Figure3—figure supplement 2; Figure 4—figure supplement 1). The number of spines/mouse/day imaged are provided in Table2 now.

4) The authors chose to only use one main methodology for their experiments- in vivo imaging using two-photon microscopy utilizing a hippocampal cranial window preparation- which is an appropriate method to measure cellular dynamics. The experiments are technically challenging. However, the authors could do more to discuss the limitations of using their techniques and put their work in the context of the literature.

We have now added a new point to the discussion termed “limitations” pointing out potential deficits of our study especially in terms of methodology. We also put our work now more in the context of the literature.

Reviewer #3 (Recommendations for the authors):The authors address an important question in the field-how do microglia in the hippocampus respond to local neuronal activity? They show that microglial motility increases in response to activity. The most convincing piece of evidence is the DREADD data, which was added in response to the comments of previous reviewers. These data are novel and will improve our understanding of memory disorders and neurodegeneration, where hippocampal microglia likely play a role.However, as previous reviewers pointed out, the experimental design is not optimal, which is not an easy problem to mitigate. The authors seem to have addressed some of the concerns by mentioning important caveats and moderating some of the claims. However, this resulted in the loss of significance of some of conclusions.Strengths:1. The new DREADD experiment addresses an important question in the field-how do microglia in the hippocampus respond to local neuronal activity? The authors now show causal evidence that microglial motility increases in response to activity. Their data are novel and will improve our understanding of memory disorders and neurodegeneration, where hippocampal microglia likely play a role (but see caveats below).2. While some of the isoflurane vs. anesthetized data are perhaps not entirely novel, they still contribute to reconciling the findings from previous publications in the field.Weaknesses:1. The window implantation may result in increased inflammation in the hippocampus, and will certainly trigger inflammatory responses in the cortex. Moreover, it is possible that inflammation is present in some mice but not others, depending on how much tissue was removed during the implantation surgery and whether hippocampal tissue was damaged.

We thank the reviewer for this important comment. Indeed, the implantation of a hippocampal window leads to initial gliosis in the hippocampus and cortex. However in a previous study we could show that this inflammatory reaction (as measured by Iba1+ and GFAP stainings) only lasts about 3 weeks after which it reseeds back to baseline levels (Gu et al. 2014). We also can exclude that hippocampal tissue was damaged. As previously shown, spine density and neuron survival in the hippocampus were unchanged between mice with and without hippocampal window implantation. Furthermore, density of cfos positive neurons upon contextual fear conditioning were unchanged between ipsi- and contralateral hemispheres.

Nevertheless, we added a sentence that addresses the possibility of a longer lasting inflammation that is independent of morphological or cell number changes as a point of consideration in the manuscript (L488)

2. The experimental timeline is another potential concern, given the issues with inflammation discussed in the previous point. Is there a reason for the 3-week interval between anesthesia, awake and TTX experiments in Figure 1? Since the interval between imaging sessions is quite long, could it be that the time from surgery affects the inflammatory response in the 3 different conditions?

There was no particular reason for the timespacing, but coordination of experiments with microscope availability and timesheduling. Since inflammation, as measured by different parameters in Gu et al. 2014, decreased already after three weeks, we did not expect any newly appearing inflammation for later time-points.

Along the same lines, could the 1-week wait time between baseline and CNO experiments in Figure 2 be excessive? Were the experiments counterbalanced so that some mice received CNO in the first session, while others received it in the second session?

We thank the reviewer for the comment. We do not think that the 1-week waiting time is excessive. The waiting time of one week was chosen primarily because of microscope booking reasons (availability of the instrument, see also point above). We started always with vehicle-injection to exclude the possibility of a long-lasting CNO-effect. Therefore, we did not start with CNO-treatment and did also not mix mice (starting with CNO or vehicle). Mixing treatment regimes would have increased variability and required a higher number of mice, which should be avoided according to 3R rules.

3. The TTX experiments are difficult to interpret due to the acute injury response from the TTX injection, which could potentially induce microglia migration in the area adjacent to the lesion.

We thank the reviewer for pointing this out. TTX was not injected, but topically applied after removal of the window. This procedure is comparable to a bath application of TTX in an electrophysiology measurement in brain slices. The removal of the window did not lead to any bleeding and the imaging region (stratum radiatum, CA1) was approximately 150µm away from the surface. During the imaging of microglia, we also did not observe a microglia process extension towards the surface, which one would have expected to observe in case of a lesion. We are therefore confident that TTX application was not accompanied by any lesion.

4. Another potential confound is the AAV-mediated over-expression of Cre recombinase, which can have toxic effects.

We thank the reviewer for raising this concern. We assume that the reviewer refers to potential neurotoxic effects of Cre-recombinase expression in neurons. However, we used for example AAV9-Cre plus a second AAV containing either an inhibiting or activating DREADD (Figure 2). If the AAV9-Cre would have had a confounding effect, why should we observe on the one hand a decrease of microglia motility (hM4D(Gi)) and on the other hand an increase in microglia motility (hM3D(Gq))? Although there might be an effect of AAV9-Cre, one would expect that both groups would be similarly affected and not in opposite directions.

5. I am not sure that the statistics in Figure 3 and Figure 4 are appropriate. The measurements from different dendrites are not truly independent observations since some were collected in the same mouse (i.e. the data is nested).

Thank you for pointing this out. We revised the statistics for Figures 3 and 4 and provide the data on individual mice now. The analyses over dendrites is now provided as supplementary figures.

Suggested approach to mitigate weaknesses:Weakness 1: The authors could discuss this caveat and quantify neuroinflammatory markers such as GFAP, microglial density and possibly microglial morphology (although from the images shown microglial do not appear to be ameboid). The non-implanted hemisphere or non-implanted tissue can be used as controls.

We thank the reviewer for the suggestion. We agree with the reviewer that microglia shown in our pictures and movies do not appear ameboid. This is the reason, why we consider that neuroinflammation was back to baseline levels at our imaging timepoints. We also did the suggested measurements in our previous work (Gu et al. 2014) and addressed the raised points above. We also provide a further supplementary figure (Figure 1 —figure supplement 2) showing ramified microglia under varying conditions.

Weakness 3: This is already addressed in the text in response to the previous reviewers but it is nevertheless a weakness that reduces the significance of the data.

We agree that this experiment has to be interpreted with care (L159-161; L304-308). Nevertheless it was the starting point for the following less invasive DREADD experiments, which could provide a more profound link between neuronal activity and microglial motility. Thereby, a potential direct affect of TTX on microglia themselves could be ruled out.

Weakness 4: Mention caveat in text

Indeed, we found publications supporting a neurotoxic effect of Cre-recombinase (Forni et al., 2006; Erben et al., 2022; Rezai Amin et al., 2019). However, these observations were made in cell cultures, during development and in dopaminergic neurons in adult mice. The latter suggesting a high viral titer and expression of Cre-recombinase as the reason for neurotoxicity in these neurons. Whether the findings from different brain regions, cells and culture systems are applicable to our experimental conditions is speculative. Furthermore, as discussed above, we observe opposite effects of inhibiting and activating DREADDs on microglia motility, while using the same AAV9-Cre. These arguments are in favor of a non-confounding effect of AAV-Cre application. We included a sentence in the limitations paragraph of the discussion to address this point.(L493497)

Weakness 6: It seems to me that a multilevel statistical model should be used (even more so since the number of dendrites imaged per mouse seems to be variable).

We thank the reviewer for the comment. The statistics for Figure 3 and 4 are now performed over mice and the statistics over dendrites is provided as supplementary (Figure3—figure supplement 2; Figure 4—figure supplement 1). Therefore, we don’t see the necessity of a multilevel statistical model.